# Geologically constrained 2-million-year-long simulations of Antarctic Ice Sheet retreat and expansion through the Pliocene

Anna Ruth W. Halberstadt [1] ✉, Edward Gasson[2], David Pollard [3], James Marschalek[4] & Robert M. DeConto [5]

Pliocene global temperatures periodically exceeded modern levels, offering insights into ice sheet sensitivity to warm climates. Ice-proximal geologic records from this period provide crucial but limited glimpses of Antarctic Ice Sheet behavior. We use an ice sheet model driven by climate model snapshots to simulate transient glacial cyclicity from 4.5 to 2.6 Ma, providing spatial and temporal context for geologic records. By evaluating model simulations against a comprehensive synthesis of geologic data, we translate the intermittent geologic record into a continuous reconstruction of Antarctic sea level contributions, revealing a dynamic ice sheet that contributed up to 25 m of glacial-interglacial sea level change. Model grounding line behavior across all major Antarctic catchments exhibits an extended period of receded ice during the mid-Pliocene, coincident with proximal geologic data around Antarctica but earlier than peak warmth in the Northern Hemisphere. Marine ice sheet collapse is triggered with 1.5 °C model subsurface ocean warming.

Based on atmospheric $CO_2$ concentrations and global temperatures, the warm Pliocene provides an analog for current and future climate and sea level[1]. However, large uncertainties hamper geologic estimates of Pliocene global sea level, and paleo shoreline reconstructions are limited in their ability to resolve the relative amplitudes and timing (hemispheric phasing) of sea-level contributions from the Antarctic and Greenland ice sheets. An upper limit on Pliocene sea level remains elusive, which propagates deep uncertainty in future sea-level projections[2]. During the Pliocene, Antarctic Ice Sheet (AIS) behavior dominated the global sea-level signal; therefore, reconstructing ice sheet dynamics during this key time period is crucial for providing context for global sea-level reconstructions, understanding glacial stability, and improving future sea level rise projections.

Previous model explorations of AIS contribution to Pliocene sea level have simulated stable ice sheet configurations under static boundary conditions[2–6]; while this approach is fairly computationally straightforward, a constant climate forcing may artificially build up (or melt) ice sheets compared to a time-evolving climate[7], and the equilibrium snapshot method can introduce additional uncertainty due to hysteresis in initial conditions[8]. This approach is also restricted to a specific time period corresponding to the specified boundary conditions; most work has focused on the mid-Piacenzian Warm Period (MPWP, 3.264–3.025 Ma)[9], but Southern Hemisphere maximum insolation occurred earlier in the Pliocene (4.23 M)[6], and other geologic proxies indicate sea-level highstands or Antarctic-proximal temperature maxima during different time intervals than the MPWP[10–12]. Previous modeling of time-evolving Pliocene AIS dynamics spans only short intervals[13] or has been inextricably tied to the benthic $\delta^{18}O$ record using inversion methods[14,15] which assumes a linear relationship between $\delta^{18}O$ and $CO_2$ as well as ice volume, though this relationship is known to be complex[16].

Here we use numerical ice sheet and climate modeling to explore ice sheet dynamics throughout the mid- and late Pliocene (4.5–2.6 Ma). Model simulations evolve transiently, reproducing unique patterns of

[1]Department of Earth and Planetary Sciences, Jackson School of Geosciences, The University of Texas at Austin, Austin, TX, USA. [2]School of Geographical Sciences, University of Bristol, Bristol, UK. [3]Earth and Environmental Systems Institute, Pennsylvania State University, University Park, PA, USA. [4]Department of Earth Science and Engineering, Imperial College London, London, UK. [5]Department of Geosciences, University of Massachusetts Amherst, Amherst, MA, USA. ✉e-mail: arhalberstadt@utexas.edu

glacial cyclicity as the ice sheet responds to variable climatic forcing driven by astronomical orbits and $CO_2$ fluctuation. We use an established ice sheet model (PSU-ISM; with hybrid ice physics using the shallow ice and shallow shelf approximations and a grounding line ice-flux formulation[17,18]). Time-varying climatic forcing is provided to the ice sheet model following the matrix method[19,20]; the appropriate climatology at each timestep is interpolated from a matrix of climate model equilibrated snapshots performed under varying $CO_2$ concentrations (285 and 421 ppm), orbital configurations (eccentricity, precession, and obliquity values characteristic of minimum, maximum, and median Antarctic summer insolation levels, at 2.967 Ma, 2.956 Ma, and 2.892 Ma, respectively), and ice sheet topographies (collapsed West Antarctic Ice Sheet with loss of East Antarctic marine basins; modern; and a Pliocene expanded glacial topography; see "Methods"). Ocean temperatures are scaled from a modern climatology using the matrix method weighting scheme to either apply a uniform ocean temperature anomaly for warmer-than-present times, or interpolate between a modern and glacial ocean for colder-than-present times ("Methods"). The matrix method interpolation can account for dynamic ice sheet changes like surface lowering, but it does not include changes to paleogeography or ocean circulation. Because climatology inputs are selected based solely on time series datasets ($CO_2$ and astronomical orbit) along with ice sheet topography at the previous timestep, this methodology is independent of the global oxygen isotope record. We develop these computational techniques in order to reconstruct and assess AIS behavior throughout the Pliocene, rather than constraining our analysis to just one extreme time interval (e.g., the MPWP). We can therefore explore the interplay of different processes at different timescales, for example, marine ice sheet margin dynamics versus precipitation across the ice sheet surface. We also explore the role of marine ice sheet and ice cliff instability feedbacks on Pliocene ice sheet dynamics; specifically, we investigate the marine ice cliff instability (MICI) mechanism that is driven by meltwater-enhanced calving processes. Two key model MICI parameters describe the propagation of water-filled crevasses (hydrofracturing) and the maximum rate of ice cliff structural failure[2,21].

Crucially, these time-evolving three-dimensional ice sheet simulations provide spatial and temporal context for geologic records. Transient model results are directly comparable to geologic records of ice sheet dynamics (for example, grounding line behavior). We compile a suite of currently available marine and terrestrial geologic data from across the Antarctic continent, synthesize these data into discrete model evaluation criteria, and systematically apply the geologic criteria to an ensemble of multimillion-year simulations performed under different combinations of key parameters (ice sheet sensitivity to ocean temperature, MICI parameterizations of ice cliff failure rates and hydrofracturing propagation, and the methodology for scaling climate input; "Methods"). Each ice sheet model simulation is compared against these datasets to identify best-fit simulations with the highest fidelity to the currently available ice-proximal geologic record. Best-fit model simulations are used to extrapolate pinpoint geologic records, disparate in space and time, into a continuous and geologically constrained reconstruction of AIS contribution to Pliocene sea level.

## Results and discussion
### Geologic records and model-data comparison
Modeled ice sheet behavior ranges widely due to key parameter variation (Fig. 1 and Supplementary Fig. S1). We first synthesize the available geologic records from across the Antarctic continent, compiling a suite of different data types, proxies, and geologic settings to validate and constrain our model simulations. This compilation is used to evaluate geologic fidelity: below we summarize model-data comparison results for (a) ice advance and retreat patterns, (b) extent of ice retreat, and (c) ice thickness changes. See "Methods" for a

comprehensive sector-by-sector description of these datasets and the specific model evaluation criteria for each sector.

Ice advance and retreat patterns are recorded by marine geophysical data and drill core sediments, illuminating the extent and frequency of Pliocene glacial expansions across the continental shelf. In the Amundsen Sea, the West Antarctic Ice Sheet (WAIS) grew out to the continental shelf break multiple times during the early and later Pliocene ($\geq 8$ and $\geq 3$ times, respectively), with a prolonged period of mid-Pliocene ice sheet retreat from about 4.2–3.2 Ma (the Pliocene Amundsen Sea Warm Period, PAWP)[22,23]. In the Ross Sea, seismic stratigraphy reveals $\geq 7$–10 episodes of widespread Plio-Pleistocene glacial advances from both the WAIS and East Antarctic Ice Sheet (EAIS)[24–26]. Although the exact ages of these unconformities remain relatively unconstrained, some of these events likely correspond to glacial erosional surfaces identified at the ANDRILL-1B site during the later Pliocene (~13 advances)[12,27]. Poor age control precludes the definite identification of a period of prolonged Ross Sea ice retreat at the same time as in the Amundsen Sea, although ANDRILL-1B paleoenvironmental reconstructions indicate an extended warm interval from 4.5 to 3.4 Ma[12], slightly earlier than the PAWP. Reconstructions of WAIS and EAIS dynamics in the Weddell Sea are extremely limited due to persistent sea ice obstructing ship access; however, the accumulation of glacially triggered debris flows on the continental shelf slope during the Pliocene suggests that the ice sheet periodically advanced to the shelf break[28,29]. Offshore of the Wilkes Subglacial Basin, sediment and drill cores indicate $\geq 12$ EAIS advances to the shelf break alternating with large-scale grounding line retreat[30–34]. Similarly, in Prydz Bay, glacial unconformities and core data reveal periodic EAIS advances across the continental shelf during the Pliocene[35–38]. In summary, these datasets reconstruct a dynamic marine ice sheet that reached the continental shelf edge during many, if not most, glacial expansions, and receded during interglacials.

These geologic criteria, with varied confidence levels based on the robustness of the geologic constraint ("Methods"), are used to evaluate our ensemble of model simulations (Fig. 2). The computational effort of performing an ensemble of multimillion-year simulations requires a relatively coarse (40 km) model spatial resolution, so our interpretation of the geologic record and model-data comparison efforts are correspondingly large-scale; however, higher-resolution nested simulations demonstrate similar patterns of grounding line fluctuation (Supplementary Fig. S2). Simulations generally reproduce orbitally paced dynamic EAIS and WAIS migration across the continental shelf during the Pliocene. However, some model members with the highest sensitivity to ocean temperature, or fastest ice cliff failure rates (maximum enhancement of MICI parameters), are not able to grow sufficiently far across the continental shelf to satisfy this set of geologic constraints. In the Ross Sea region, only those simulations with lower sensitivity to ocean temperatures advance all the way to the shelf break as indicated by the geologic record. Simulations with lower sensitivity to ocean temperature and less extreme MICI parameters are able to reproduce the observed patterns of ice sheet fluctuation.

Most model simulations reconstruct a long period of ice sheet retreat during the PAWP in all catchment regions (not just the Amundsen Sea). This modeled warm interval is therefore slightly offset from the Ross Sea and Prydz Bay geologic records, with prolonged ice recession from ~4.1 to 3.2 Ma (rather than 4.5–3.4 Ma as in ref. 12, or 4.6–4.0 Ma as in ref. 38). Also, model ensemble members that generally produce sufficient glacial expansions across the Ross Sea continental shelf during the later Pliocene also advance during the early Pliocene, although direct geologic evidence for glacial expansion during that time is absent[12,27].

Constraining the extent of past ice sheet retreat beyond the modern configuration requires more indirect geologic datasets. During the Pliocene, large-scale ice sheet collapse events are recorded by

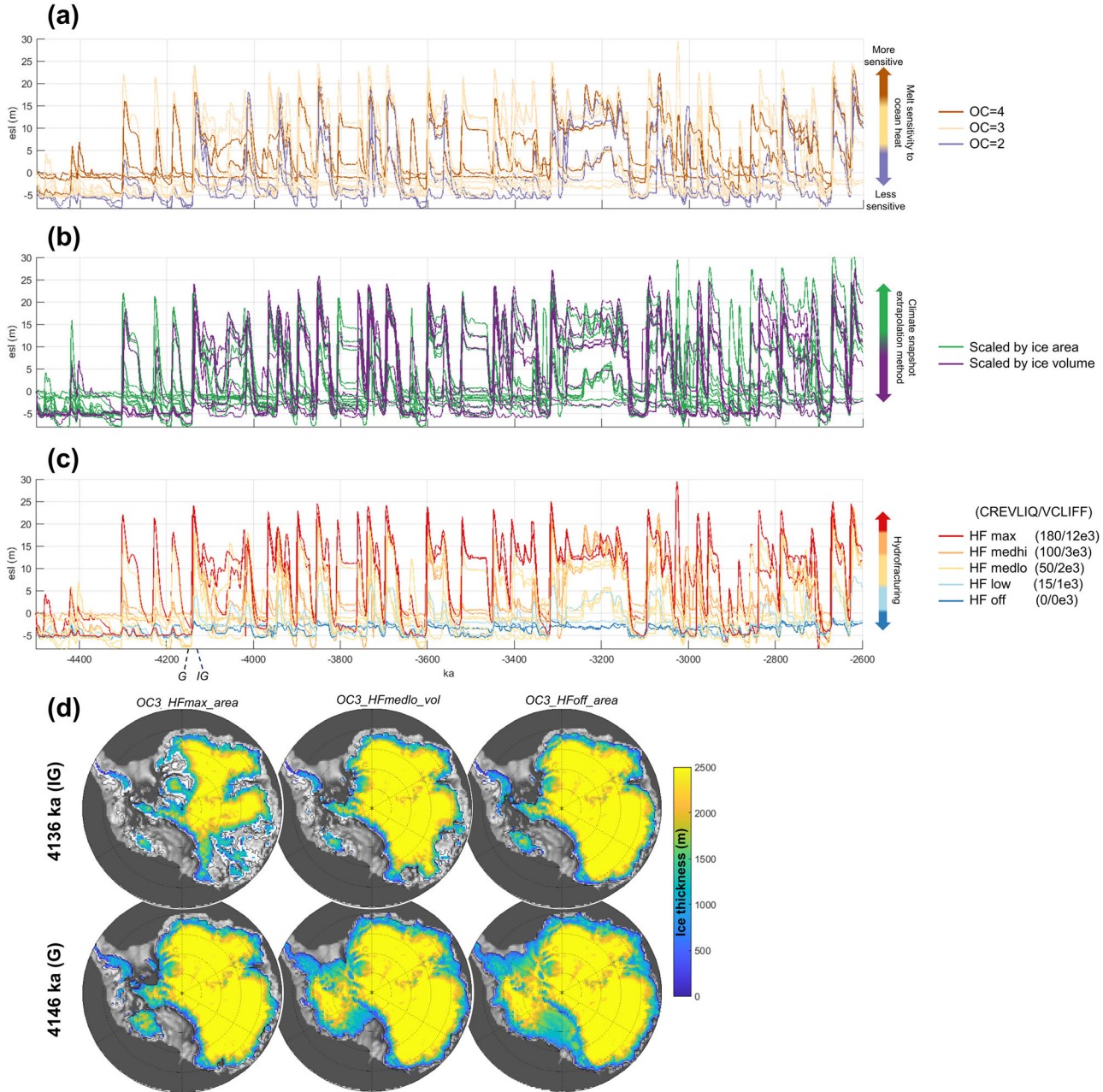

**Fig. 1 | Pliocene model ensemble results.** An ensemble of ice sheet model simulations, performed under three main parameter variations, produces a range of glacial behavior across glacial/interglacial cycles. **a–c** Simulated Pliocene Antarctic esl (equivalent global mean sea-level contribution relative to modern, calculated from the total ice amount in the domain divided by global ocean area). The full model ensemble is shown in (**a–c**), but model runs are colored by different parameter values: **a** sensitivity to ocean temperature, **b** method of scaling the climate forcing methodology, and **c** hydrofracturing (see the text for parameter descriptions). **d** Grounded ice sheet configurations for representative interglacial (IG) and glacial (G) time slices for select model members demonstrate the wide range in spatial variability that can be simulated under different sets of parameter values.

iceberg-rafted debris accumulation rates and sediment provenance analyses, as well as inland outcrops of open-marine sediments. Specifically, far-traveled iceberg-rafted debris pulses are attributed to destabilization events of large-scale ice collapse in the Wilkes Subglacial Basin and Aurora Subglacial Basin under warmer-than-present conditions[39,40], suggesting significant grounding line retreat into these subglacial basins. Marine diatoms in Transantarctic Mountain outcrops[41] have also been interpreted as indicators of ice collapse over Aurora and Wilkes subglacial basins[42,43]. Further evidence for inland erosion is provided by offshore records of terrigenous sediments[32,44]. The inland extent of retreat across Wilkes Subglacial Basin during

these collapse events can be constrained by (a) $\varepsilon_{Nd}$ measurements of sediments that were eroded from geochemically distinct regions of bedrock and transported offshore[45], suggesting that grounding line retreat never entered an inland source region; and (b) low cosmogenic nuclide concentrations in ANDRILL-1B sediments which preclude land exposure of much of the Transantarctic Mountain region and the southernmost part of the Wilkes Subglacial Basin[46]. In Prydz Bay, large-scale EAIS retreat is also indicated by inland outcrops of open-marine sediments that were deposited during periods of grounding line and ice shelf retreat by hundreds of kilometers[47,48]. In Aurora Subglacial Basin, however, model-data comparison is complicated by directly

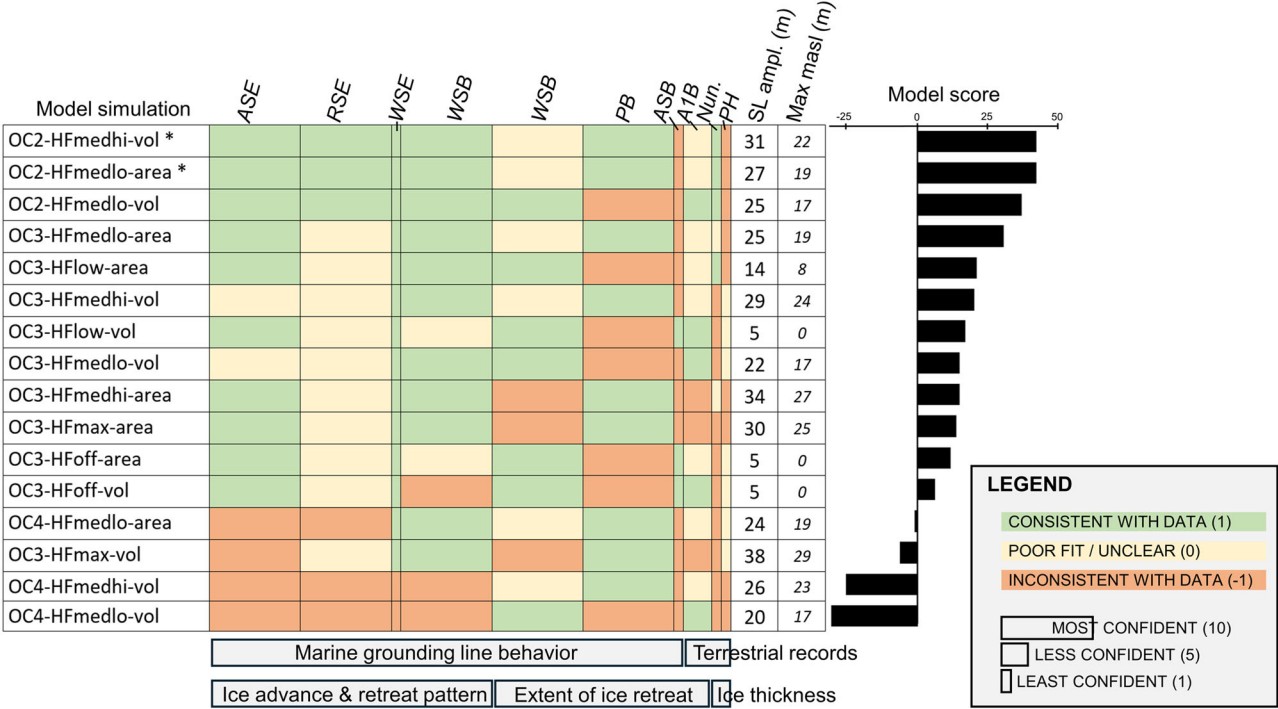

**Fig. 2 | Model-data evaluation.** Each model simulation is assessed for consistency with the available geologic data across marine sectors and terrestrial constraints (see Supplementary Table S1 for specific geologic criteria). ASE Amundsen Sea Embayment, RSE Ross Sea Embayment, WSE Weddell Sea Embayment, WSB Wilkes Subglacial Basin, PB Prydz Bay, ASB Aurora Subglacial Basin, A1B ANDRILL-1B provenance, Nun. interior nunataks, PH Pirrit Hills. Cell color reflects the evaluation of model-data consistency, and cell width indicates the confidence of this evaluation; a 'least confident' classification may result from the equivocal nature of geologic evidence, or inherent difficulties with comparing model results with that particular kind of geologic data (e.g., spatial resolution). Model scores are calculated by multiplying the model-data agreement score for each criterion (1, 0, or −1)

by the confidence weight (10, 5, or 1), summed across columns. Model member naming convention reflects the parameter combination (ocean temperature sensitivity OC 2,3,4−hydrofracture parameterization HF off,low,medlo,medhi,max−climate matrix scaling approach area,vol). Total sea-level amplitude "SL ampl." reports the largest difference in sea-level equivalent (m SLE) between maximum and minimum ice sheet configurations, while "Max masl" reports the maximum sea-level equivalent contribution above present. Geologic data from Wilkes Subglacial Basin is split into two categories: datasets constraining glacial advance and retreat across the continental shelf, versus datasets constraining the inland extent of grounding line retreat. Asterisks denote the two best-fit model runs, identified by weighting simulations based on model-data comparison confidence.

conflicting data-based interpretations: geophysical evidence suggests that grounding line retreat across the Aurora Subglacial Basin was limited to ~150 km inland from its modern position[49], but iceberg-rafted debris pulses likely originated from larger-scale retreat in this region[39,40].

Model members with little or no ice cliff failure (MICI parameters set to zero or low) do not produce sufficient grounding line retreat to satisfy the geologic evidence for large-scale grounding line retreat across Wilkes Land continental shelf or into the Wilkes Subglacial Basin. Model ensemble members with zero or low MICI parameterizations also do not drive enough ice sheet and ice shelf recession in Prydz Bay to simulate periodic open-marine environments occurring upstream of the glacially reworked diatomaceous sediment outcrops. However, model simulations with very high parameterized MICI sensitivity produce frequent ice sheet retreat into a geologically contraindicated inland source region[45,46]. Only simulations with intermediate MICI parameterizations are consistent with the evidence for ice sheet retreat across Wilkes Subglacial Basin as well as the Prydz Bay geologic record.

Past ice thickness changes can be reconstructed from cosmogenic nuclides measured at exposed mountain peaks. Although these terrestrial data from the Pliocene are extremely limited, Yamane et al.[50] report episodes of interior Pliocene ice sheet thickening at various Antarctic nunataks. Their observations of ice sheet thickening are consistent only with models that have lower parameterized sensitivity to ocean temperatures: in these simulations, inland thickening occurs during interglacials due to precipitation, while coastal thickening

occurs during glacials due to marine ice growth. Halberstadt et al.[51] also used cosmogenic nuclide exposure ages to characterize the fraction of time that each elevation along a mountain peak has spent ice-covered, thus reconstructing the frequency behavior of ice sheet thinning and thickening. This approach has only been employed at the Pirrit Hills; however, the pattern of cyclic bedrock exposure at this location is not consistent with any model simulations. Model-data comparison using exposure age datasets is hampered by the coarse model resolution, which does not resolve the mountain peaks where data were collected; additionally, at the Pirrit Hills site, the ice thickness frequency dataset is integrated across a different time period as the simulations in our model ensemble.

This synthesis of Pliocene geologic data reconstructs a dynamic marine ice sheet that grew across the Antarctic continental shelf during glacial periods and retreated beyond the modern configuration during interglacials (with evidence of periodic large-scale ice sheet collapse). These detailed datasets are leveraged as sector-by-sector model evaluation criteria ("Methods"; Supplementary Table S1) and used to narrow down the full model ensemble (Fig. 1) to identify the most geologically consistent simulations (Fig. 3). In accordance with the geologic record, best-fit model simulations reproduce glacial periods of ice sheet expansion to the continental shelf edge, with episodic ice sheet retreat deep into EAIS marine basins (Fig. 3d, e).

## AIS contribution to Pliocene global mean sea level

Global mean sea-level (GMSL) highstands during the Pliocene remain poorly constrained; large uncertainties plague benthic $\delta^{18}O$

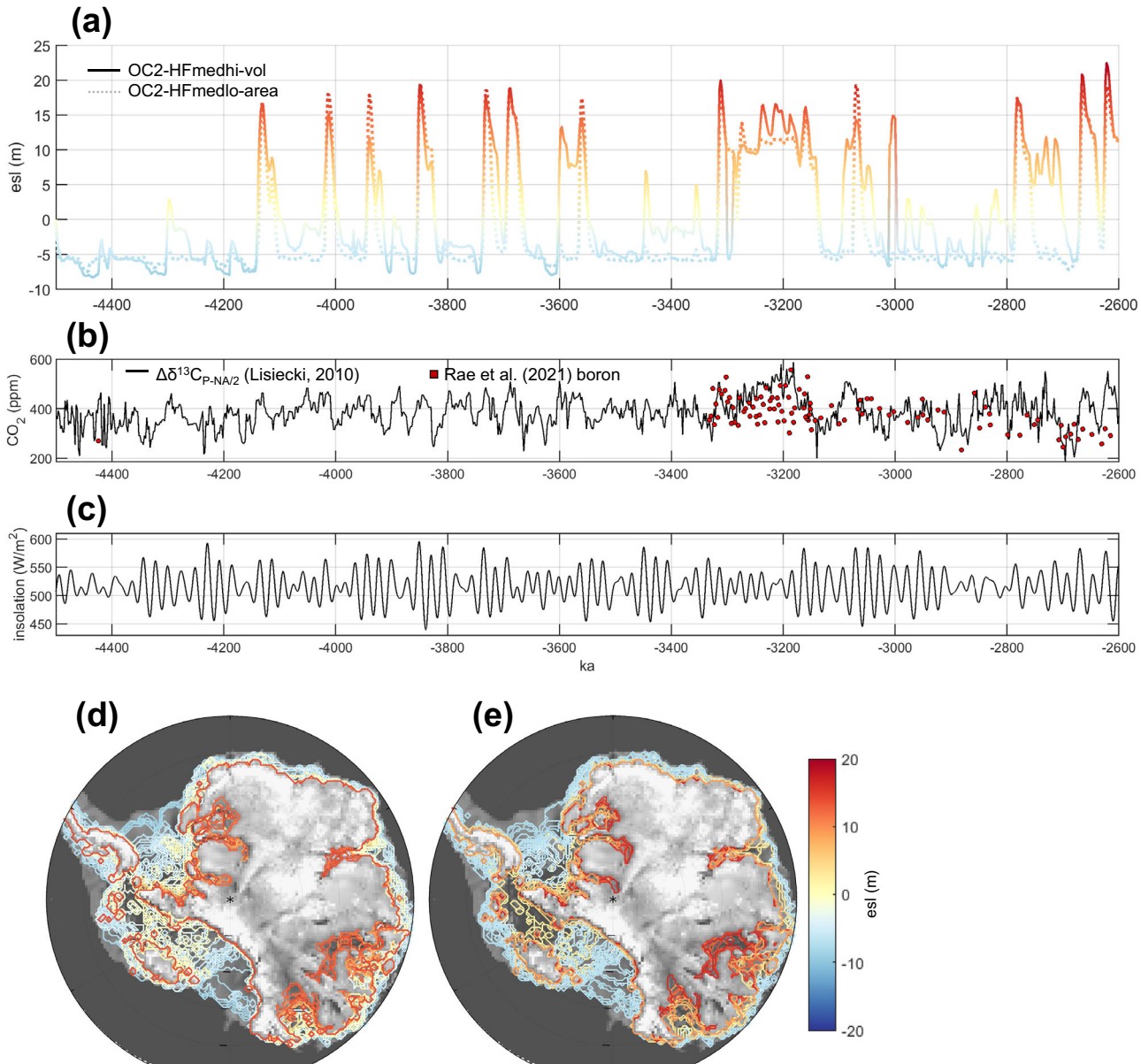

**Fig. 3 | Antarctic contribution to Pliocene sea level in best-fit model simulations. a** Best-fit model runs (see Fig. 2 for model-data evaluation) simulate large-scale Antarctic ice sheet fluctuations of up to -20 m esl (contribution to global sea level). **b** Orbitally resolved $CO_2$ proxy cf. ref. 103, constrained by boron $CO_2$ reconstructions. **c** January insolation anomaly at 80°S. Grounding line positions through time are plotted for best-fit runs **d** OC2-HFmedhi-vol and **e** OC2-HFmedlo-area, showing the spatial variability of the ice sheet ranging from grounded ice expansion across the continental shelf to collapse of all marine-based ice.

reconstructions of sea level[16,52], although far-field geologic records imply a sea-level contribution from the AIS of >10 m[10,53,54]. Pliocene GMSL records provide basic constraints on ice sheet dynamics and global climate during past warm periods but cannot directly deconvolve sea-level contributions from Antarctica versus Northern Hemisphere sources. If the Greenland and Antarctic ice sheet fluctuations were antiphased, Greenland ice sheet growth could have masked contemporaneous large-scale AIS mass loss, and future sea-level projections constrained by Pliocene GMSL constraints will underestimate the AIS contribution. Model simulations of transient Antarctic ice sheet evolution can therefore provide key context for interpreting GMSL records with respect to ice sheet stability.

The two model runs that are most consistent with the geologic record simulate glacial-interglacial ice sheet changes on the order of 25 m of equivalent sea level (esl) from Antarctica (Fig. 3; calculated as the difference between minimum and maximum configurations across

the simulation), with highstands around 18 m esl above present. If we assume that the Greenland ice sheet contributed up to ~5–7 m sea-level-equivalent during the Pliocene and deglaciated out of phase with the AIS (as suggested by refs. 3,13), then our model ice sheet fluctuations would result in GMSL amplitudes of up to ~18 m. With these assumptions, GMSL ranges from +18 m during an Antarctic retreat/Greenland growth period ( + 18 m from the Antarctic Ice Sheet plus 0 m from an approximately modern-size Greenland) to -0m during an Antarctic growth/Greenland retreat (−7 m from Antarctica plus +7 m from Greenland), resulting in GMSL amplitude of 18 m. These calculations assume that the only Northern Hemisphere source of ice was Greenland, although future work will investigate potential Northern Hemisphere ice sheet growth. This estimate of 18 m GMSL variation is consistent with a geologic reconstruction of Pliocene GMSL amplitudes of up to 25 m from a continuous water-depth proxy[53]. If ice sheets in both hemispheres advanced at the same times (in phase), our

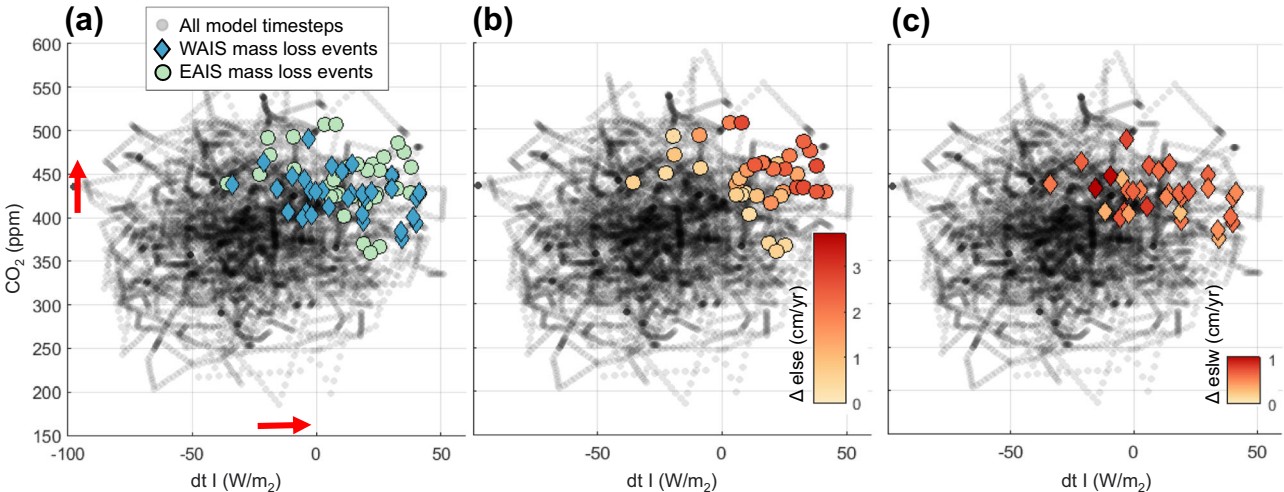

**Fig. 4 | $CO_2$ concentrations and insolation values corresponding to mass loss events.** $CO_2$ concentrations and summer insolation values corresponding to each modeled collapse event (defined as the onset of large-scale ice sheet mass loss, with rates exceeding 1 m/kyr) are compared to the range of forcing values across all model timesteps. Insolation (dt I) is reported as a January 80°S insolation anomaly from modern. **a** Ice sheet mass loss events are separated by East Antarctic Ice Sheet (EAIS) and West Antarctic Ice Sheet (WAIS) domains. WAIS mass loss events are characterized by the disappearance of marine-based portions of the ice sheet; EAIS mass loss events are characterized by grounding line retreat deep into EAIS marine basins. **b** EAIS and **c** WAIS mass loss events are colored by rate of change (sea-level equivalent, in cm/yr).

model ice sheet fluctuations would produce GMSL variability of up to 32 m. This modeled variability exceeds the direct water-depth reconstruction[53]; it falls within the range of $\delta^{18}O$-derived sea-level reconstructions[55–59], though $\delta^{18}O$ estimates of sea level have significant uncertainties[16].

The simulated sea-level amplitudes and model scores within our ensemble are not directly correlated; some of the worst-fit simulations also have large sea-level fluctuations (Fig. 2). Thus, our compilation of geologic data can inform not just the AIS contribution to Pliocene sea level, but also resolve the pattern of past ice sheet dynamics. This highlights the added value of considering ice-proximal data along with a sea-level constraint when evaluating past AIS simulations.

We also note that our best-fit modeled AIS sea-level contributions are negative (i.e., larger than today) during part of the early Pliocene. Despite larger ice volumes, we simulate WAIS and even EAIS grounding line retreat (Fig. 3 and Supplementary Fig. S1) as precipitation outweighs interglacial marine mass loss under this early Pliocene combination of relatively low $CO_2$ proxy values and invariant insolation.

## Spatial extent of interglacial ice sheet retreat

The ability of models to reproduce ice sheet retreat in the Pliocene is of key importance to future sea-level projections[60,61]. Various modeling groups have developed numerical schemes to produce the amount of ice loss generally indicated by paleo sea-level records; for example, sub-grid ocean melting[6], basal sliding[62], or MICI mechanisms (hydrofracturing and ice cliff failure)[21]. For all model approaches, constraining these parameterizations is critical for past and future simulations; Pliocene GMSL targets have been used to calibrate model ice sheet parameters (e.g., ref. 2), but this approach is limited by a lack of spatial information regarding the locations of large-scale ice mass loss.

MICI is based on physical theory[63] but the onset and details of these processes remain uncertain[64–66] and continue to be debated[67]. Because MICI mechanisms are activated under an abundance of surface meltwater, the geologic record of ice sheet behavior during the warm Pliocene provides an important validation of the large-scale impact of these processes. Here we use the spatial constraints from our compilation of ice-proximal Antarctic geologic records to eliminate extreme end members of the MICI parameter combinations explored by refs. 2,60, showing that only intermediate values are geologically

consistent. Specifically, zero or low MICI parameter values cannot generate enough grounding line retreat across Wilkes Subglacial Basin (to produce large pulses of far-traveled iceberg-rafted debris) or Prydz Bay (to deposit inland open-marine sediments), but maximum values prevent sufficient glacial expansion across continental shelves (e.g., in the Amundsen Sea) and drive too much retreat into Wilkes Subglacial Basin (eroding into the Adelie craton region, and also exposing terrestrial sediments in the ANDRILL catchment region). Rapid rates of MICI-driven collapse are also indicated by a geologic record of iceberg calving used to infer inland retreat of the ice sheet margin across Wilkes Subglacial Basin on the order of a few thousand years[44].

Unlike DeConto et al.[2], we did not systematically sample the MICI parameter space, and our simulations use a different climate forcing methodology and vary other parameters, so we do not directly constrain their future sea-level projections here. However, our model-data comparison using spatial geologic information provides an upper and lower limit on geologically consistent MICI parameter values (Supplementary Fig. S3), although the exact values identified here are specific to this ice sheet and climate model setup with associated uncertainties. The intermediate MICI values that are required to satisfy the compiled Pliocene geologic record also produce ice sheet recession into EAIS marine basins in the future.

## Modeled thresholds for ice sheet collapse

Ice sheet fluctuations are driven by variation in climate forcing, acting in tandem with internal feedbacks. At million-year timescales, $CO_2$ plays a dominant role on ice sheet volume[68]; at glacial/interglacial timescales of 10–100 kyr, fluctuations of both $CO_2$ and insolation drive glacial cyclicity[69,70], although the thresholds and internal feedback mechanisms governing ice sheet stability remain elusive[71,72].

In our numerical experiment, modeled ice sheets are sensitive to both $CO_2$ and summer insolation. Figure 4 highlights the $CO_2$ concentrations and insolation values associated with each ice sheet mass loss "collapse" event (based on an ad-hoc criterion to identify the onset of mass loss rates exceeding 1 m/kyr). The specific thresholds of these forcings vary for each simulated collapse event and are generally characterized as >400 ppm $CO_2$ and >-20 W/m² in our simulations (Fig. 4a), although the precise values of these thresholds are model-specific and depend on climate model sensitivity and model

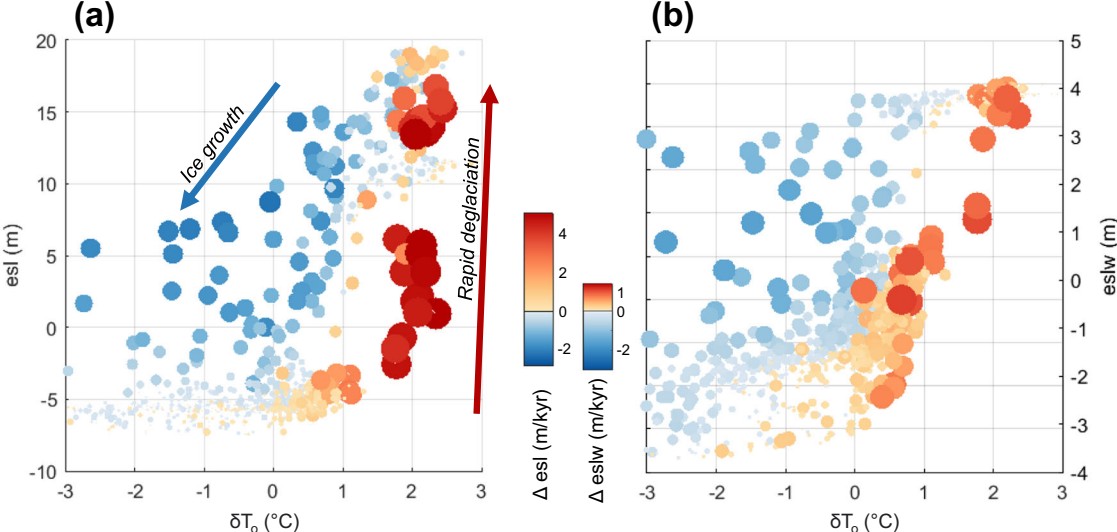

**Fig. 5 | Thresholds and rates of ice sheet change.** Each point represents one time slice throughout the simulation (best-fit run OC2-HFmedlo-area), with color and size of the plotted points representing the rate of ice sheet change (in m sea-level equivalent/kyr) at each timestep. Ice sheet deglaciation rates (red) are much faster than growth rates (blue) for both the (**a**) total Antarctic Ice Sheet and **b** West

Antarctic Ice Sheet. The subsurface ocean temperature scaling to the ice sheet model ($\delta T_o$) is calculated at every timestep using the model matrix weighting scheme, which considers variations in $CO_2$, insolation, and ice sheet configuration (Supplementary Fig. S7).

parameterizations. Note that insolation values are recorded as an anomaly from modern (specifically, at January 80°S latitude), and many model ice sheet mass loss events occur at negative values (i.e., less summer insolation than today), though this weaker insolation forcing requires a higher-than-modern $CO_2$ concentrations to generate ice sheet collapse. The simulated collapse of EAIS marine basins is driven by slightly stronger forcings compared to WAIS collapse; a stronger combined forcing is required to trigger MICI in EAIS sub-glacial basins.

For ice sheet retreat into deep EAIS marine basins, the rate of mass loss is proportional to the strength of the combined greenhouse gas and orbital forcing (Fig. 4b). In our model, $CO_2$ and insolation work together to produce warm subsurface ocean temperatures (driving melt and ice sheet recession at marine grounding lines) along with surface meltwater (which enhances surface crevassing, ice shelf loss, and ice cliff calving rates). Unlike the EAIS, WAIS modeled mass loss rates are not clearly proportional to the total strength of forcings (Fig. 4c). This suggests that Pliocene WAIS collapses were not domi-nated by one clear forcing mechanism or threshold. For example, some WAIS collapses could have been triggered by warmer ocean temperatures while others were driven by surface melt and hydro-fracturing. Another possible explanation is that the strength of the forcing for many individual WAIS collapse time intervals greatly exceeded the necessary threshold $CO_2$ concentration or insolation level, obscuring a clear signal of threshold values in Fig. 4c. Ice shee-t hysteresis and surface mass balance patterns could have also affected the unique stability of each interglacial WAIS configuration, precluding a clear relationship between forcing strength and ice sheet response.

In our simulations, the fastest episodes of Antarctic ice loss are triggered when subsurface ocean temperatures warm more than 1.5 °C (Fig. 5a); however, this signal is dominated by EAIS mass loss. WAIS tipping points occur under a wider range of temperature anomalies (about 0–1 °C; Fig. 5b), also suggesting a wider range of interacting mechanisms contributing to collapse. In both scenarios, ice sheet regrowth from a collapsed state begins across a much wider range of forcings (1–3 °C ocean temperature anomaly, which roughly occurs under ~350–500 ppm $CO_2$ in our matrix climate scaling approach).

Figure 5 also highlights the difference in rates of ice growth versus ice loss; modeled ice sheet growth generally occurs more slowly as ice shelf pinning points coalesce, while deglaciation is characterized by rapid ice sheet collapse driven by marine ice sheet (and ice cliff) instabilities.

## Mass loss at marine margins versus increased surface accumulation

Pliocene warmth drove grounding line retreat at marine margins, but also increased the amount of precipitation reaching the interior ice sheet surface; these competing processes have both been invoked in future ice sheet and sea-level projections[73,74]. Geologic records, especially in Wilkes Subglacial Basin, indicate episodes of large-scale ice mass loss[39]; however, the Pliocene ice sheet also periodically thick-ened, as recorded by long-term exposure ages from nunataks (ice thickening at the measured sites ranged from 150 to 800 m)[50]. This suite of mountain-peak measurements can provide data "anchors" for reconstructing past ice sheet thickness changes; numerical models contextualize these local measurements in space and time.

Using our transient model simulations, we evaluate if ice sheet thickening at these nunataks occurred during cold glacial periods (i.e., thickening was due to ice sheet growth at marine margins, amplifying GMSL lowstands) or warm interglacial periods (i.e., thickening was due to increased surface accumulation, countering GMSL highstands). We find that the nunataks located in the ice sheet interior (e.g., ~300 km inland; Area 1 in ref. 50) are isolated from marine drawdown effects (Fig. 6a, b); thickening is antiphase with total ice volume (Fig. 6c) suggesting that these locations are mostly influenced by increased precipitation during warm periods. Precipitation rates across the continent are greater in our climate model with a "warm interglacial" astronomical orbit compared to the "cold glacial" orbit (Fig. 6d). Although increased precipitation during warm periods influences both the margin and interior of the ice sheet, at coastal sites (e.g., ~100 km inland or less; Area 2 in ref. 50), model ice sheet thick-ness changes occur in phase with total ice volume fluctuations. This results from dynamic drawdown of marine-based ice, which propa-gates inland and influences terrestrial ice thicknesses. Indeed, both coastal sites from ref. 50 fall within the zone of influence from marine

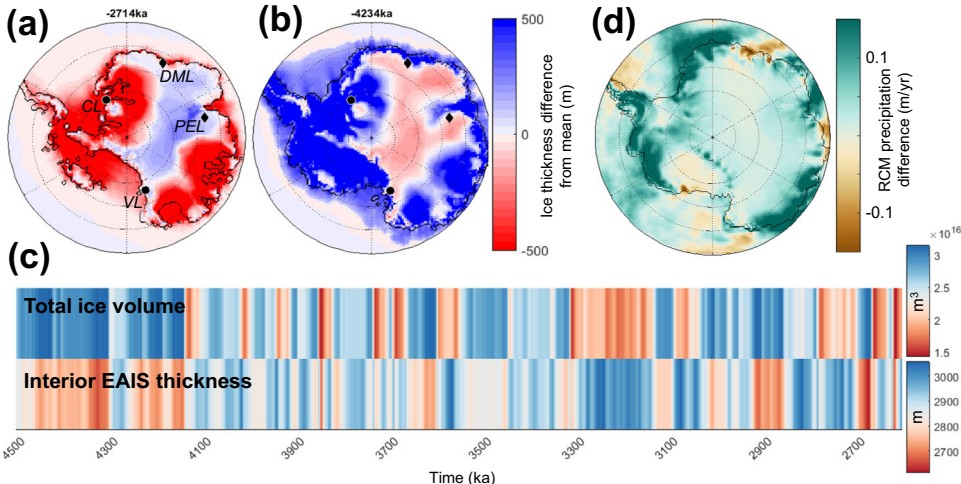

**Fig. 6 | Precipitation-driven thickening in the ice sheet interior versus dynamic drawdown at the margins.** Ice thickness deviation from the simulation ice thickness mean is shown during an interglacial (**a**) and glacial (**b**). During interglacials (**a**), ice sheet drawdown from marine margins impacts coastal areas, and increased precipitation drives interior thickening (vice versa for glacials; **b**). The locations of nunataks from Yamane et al.[50] are shown in (**a**, **b**); Area 1 sites are more inland and denoted by black diamonds (DML: Dronning Maud Land–Sør Rondane Mountains, PEL: Princess Elizabeth Land–Grove Mountains) and Area 2 sites are more coastal and denoted by black circles (CL: Coats Land–Shackleton Range, VL: Victoria Land

–Dry Valleys). **c** Total grounded ice sheet volume compared to a time series of East Antarctic Ice Sheet (EAIS) interior thickness (80–90°S, 60–120°E) shows an anti-phased relationship. **a**–**c** Model output shown for the best-fit run OC2-HFmedhi-vol. **d** Interior precipitation rates are greater in the climate model with a "warm-interglacial" astronomical orbit, compared to the "cold-glacial" orbit (plot shows the difference in precipitation rate between climate model snapshots with 421 ppm $CO_2$, a modern ice sheet topography, and glacial vs interglacial astronomical orbits).

margins (Fig. 6a, b; the Dry Valleys site is located at the very edge of this marine drawdown zone).

Our simulations provide temporal context for the various locations where Yamane et al.[50] reconstruct higher-than-present ice elevations: thickening at their coastal sites occurred during Pliocene glacials while thickening at their interior sites occurred during Pliocene interglacials, similar to ice thickness changes in the late Quaternary. Despite increased surface accumulation during warm periods, however, interglacial ice sheet contribution to GMSL was overwhelmingly dominated by mass loss at marine margins.

### Antarctic mid-Pliocene warmth

Transient model results indicate a prolonged period of mid-Pliocene Antarctic Ice Sheet recession in agreement with ice-proximal geologic records. Seismic surveys and drill core records of ice dynamics in the Amundsen Sea reconstruct an extended period of ice sheet retreat spanning multiple glacial/interglacial cycles, with only a few sporadic grounding line advances to the mid and outer shelf (the PAWP; 4.2–3.2 Ma)[22,23]. Our modeled grounding line behavior is highly accordant with this reconstruction; the simulated ice sheet in the Amundsen Sea grows out across the continental shelf repeatedly during glacial periods in the early and late Pliocene but remained mostly receded during this warm interval (Fig. 7, top row). In fact, model grounding line behavior in all major Antarctic catchments indicates a prolonged period of receded ice during this time, which also generally corresponds to the timing of greatest warmth in the Ross Sea ANDRILL record[12] (Fig. 7), as well as reconstructions of prolonged sea surface temperature increase around the Antarctic margin (ref. 11; although elevated SSTs in Prydz Bay also occurred earlier than our modeled warm period[75]).

The MPWP (3.264–3.025 Ma) has been a primary focus of the paleoclimate community, with SSTs ~3 °C above present at times[76]; however, most of the reconstructed warm SST anomalies (and, indeed, most of the available data) are concentrated in the Northern Hemisphere[76–78]. Our model results presented here, along with recent geologic evidence for ice recession from 4.2 to 3.2 Ma, suggest an earlier period of extended Antarctic-wide ice sheet recession from the

continental shelf, notably consistent with a 4.23 Ma Antarctic insolation maximum[6]. This earlier Antarctic warm period occurred mostly before the MPWP but later than the Pliocene Climatic Optimum (4.4–4.0 Ma)[10], suggesting that the Northern and Southern hemispheres may have had different periods of peak Pliocene warmth with divergent timing of maximum GMSL contributions from Antarctica and Greenland ice sheets.

Here we compile a range of Pliocene ice-proximal geologic records across the Antarctic continent to constrain and validate multimillion-year transient ice sheet model simulations. In accordance with the suite of available geologic data, our best-fit models reconstruct a dynamic marine ice sheet that grew across the Antarctic continental shelf during glacial periods and retreated beyond the modern configuration during interglacials on orbital timescales. Model simulations with the highest fidelity to the geologic record produce high-amplitude (~25 m) GMSL contributions from Antarctica throughout the Pliocene. Our simulations are consistent with independent geologic reconstructions of mid-Pliocene GMSL amplitudes[53]. Model grounding line behavior across major Antarctic catchments indicates a period of prolonged ice sheet recession during the mid-Pliocene, coincident with proximal geologic data around Antarctica but earlier than the MPWP. Our model-data comparison indicates that only intermediate values of modeled MICI parameters are consistent with the geologic record, which can help to constrain future sea-level projections. The onset of rapid deglaciation, leading to marine ice sheet collapse, is triggered with 1.5 °C model subsurface ocean warming.

Here we integrate an extensive compilation of geologic data with physically based numerical modeling to reconstruct an Antarctic Ice Sheet that was highly sensitive to Pliocene warmth. This reconstruction supports projections of ice shelf loss and ice sheet collapse[79,80] as future air and ocean temperatures approach Pliocene interglacial levels.

## Methods

### Ice sheet modeling

We conduct transient Pliocene model simulations (4.5–2.6 Ma) of the AIS using the PSU-ISM[17], a hybrid shallow ice/shallow shelf

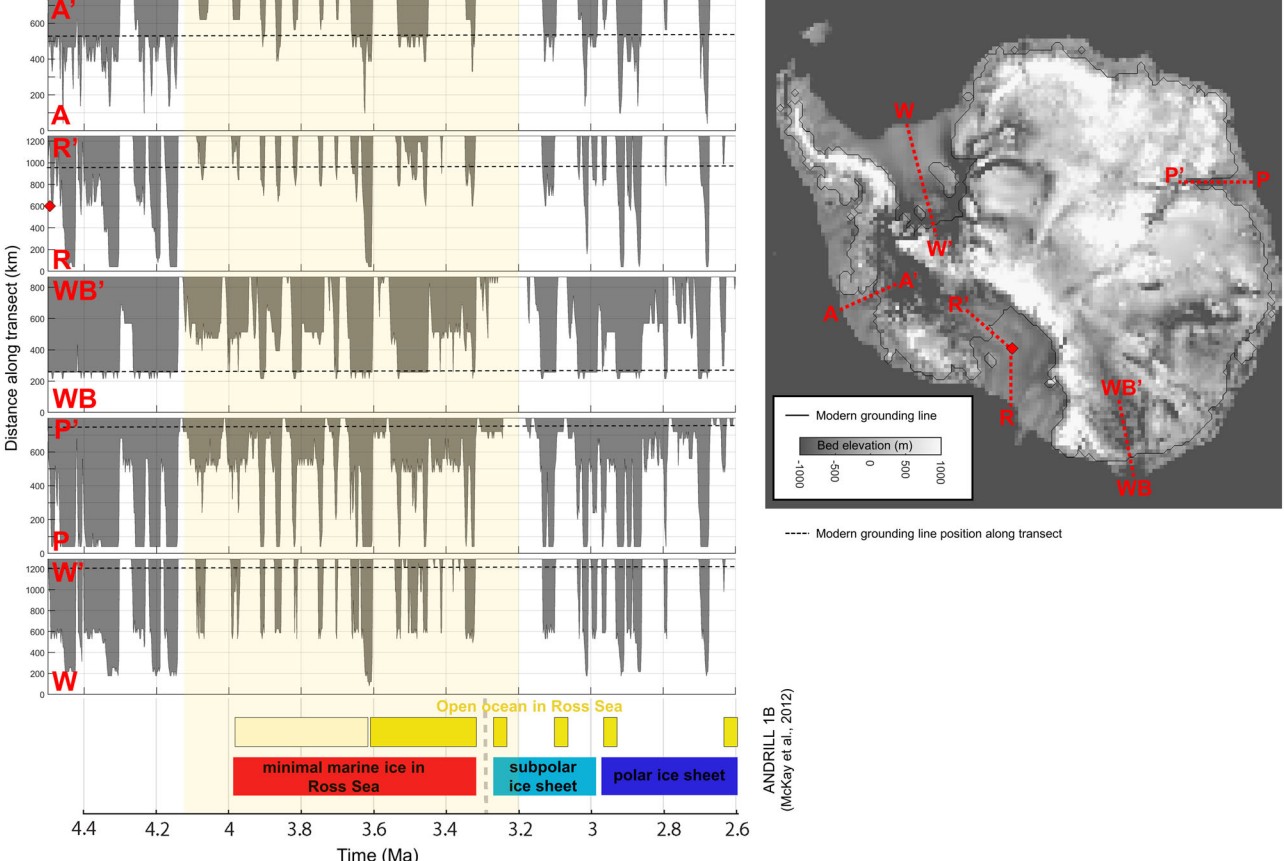

**Fig. 7 | Modeled interval of prolonged mid-Pliocene ice sheet recession in all major catchments.** Model grounding line position (OC2-HFmedhi-vol) in each major Antarctic catchment is plotted along a continental shelf transect indicated in the spatial plot. The modern (modeled) grounding line in each catchment region is denoted by the black dashed horizontal line. Yellow shading indicates the "Pliocene Amundsen Sea Warm Period" (PAWP), a time when the ice sheet was primarily

receded in the Amundsen Sea[22] (4.2–3.2 Ma, although the modeled warm period begins a bit later at ~4.1 Ma). A similar period of prolonged mid-Pliocene ice sheet recession is evident in other catchments around the Antarctic Ice Sheet during this time. This time period coincides with the time of greatest warmth in the Ross Sea ANDRILL-1B record[12].

approximation ice sheet model (ISM) with a grounding line ice-flux formulation[18] that demonstrates grounding line reversibility and reproduces theoretical and full-Stokes modeled grounding line behavior in idealized model intercomparison studies[81–83]. Model simulations are conducted as in ref. [2], with the description of additional techniques below.

Three key model parameters are systematically varied within plausible ranges to produce an ensemble of simulations. (1) Ice sheet sensitivity to ocean temperature (OCFACMULT = 2, 3, 4) is a dimensionless coefficient multiplying the sub-ice basal ice melt rates (parameterized as a quadratic dependence on temperature following ref. [84]). (2) Marine ice cliff instability parameters (VCLIFF = 0, 1, 2, 3, 12, and CALVLIQ = 0, 15, 50, 100, 180) describe the maximum rate of cliff wastage rate horizontally into the ice edge due to cliff failure (VCLIFF; $km\,yr^{-1}$; for context, terminus retreat velocities at Jakobshavn Isbrae have been measured up to ~12 $km\,yr^{-1}$ (see ref. [85])), and the enhancement of surface crevassing with increasing rain and surface meltwater availability (CALVLIQ; $m^{-1}\,yr^2$). VCLIFF and CALVLIQ parameter combinations range from inactive 'HFoff' to the maximum value "HFmax" tested in ref. [60]. (3) The matrix methodology for extrapolating the ice-extent weight was based on either the total grounded volume or the grounded area of the ice sheet. In other words, the ice-extent weight applied to each snapshot climatology is evaluated based on how closely the grounded ice volume (area) of the preceding time slice matches the total grounded ice volume (area) of the ice sheet topography that was used to produce that climatology. Each parameter

combination produces a reasonable ice sheet configuration under preindustrial (Supplementary Fig. S4) and Last Glacial Maximum boundary conditions. Surface mass balance is calculated using a positive-degree-day scheme, with a lapse-rate correction for topographic differences. Eustatic sea level is kept at zero; we find minimal model sensitivity to eustatic sea-level variations up to 60 m scaled by Northern Hemisphere insolation. For the computational efficiency necessary for multimillion-year simulations, we use a 40 km grid resolution; higher-resolution (15 km) nested simulations produce similar grounding line fluctuation, indicating that our results are not biased by the necessarily coarse resolution of our continental ensemble (Supplementary Fig. S2).

Initial conditions for our simulations are provided by a modern ice sheet configuration[2]. Pliocene paleotopographic reconstructions deviate only slightly from modern[86,87]; sensitivity tests conducted using paleotopography produce similar modeled ice sheet fluctuations, but with slightly reduced glacial/interglacial variability, potentially due to the slightly shallower Pliocene subglacial bathymetry and coarser resolution and smoother bed topography of the paleotopographic reconstruction.

At each timestep, unique temperature and precipitation fields are provided to the ISM from a matrix of 18 climate model simulations, following the matrix method[19,20]. Climate model "snapshots" in the matrix were produced at two levels of atmospheric $CO_2$ (285 and 421 ppm; Supplementary Fig. S5a, based on ref. [88]), three orbital configurations from Pliocene time periods with minimum, maximum, and

median Southern Hemisphere summer insolation (2.967 Ma, 2.956 Ma, and 2.892 Ma, respectively; Supplementary Fig. S5b), and three ice sheet topographies (collapsed West Antarctic Ice Sheet and loss of East Antarctic marine basins, modern, and a representative Pliocene expanded glacial topography; Supplementary Fig. S5c). For each matrix model member, a global atmospheric circulation model with a slab ocean (GENESIS GCM)[89] was equilibrated under a set of unique boundary conditions. In the GCM, the vertical heat flux from the ocean to the base of sea ice is iteratively set to be proportional to 50–60°S slab ocean temperatures within the range of previously validated values. GCM output was then dynamically downscaled to a 60 km resolution over the Antarctic Ice Sheet using a regional climate model (RegCM3)[90] to provide the temperature and precipitation fields passed to the ice sheet model (with temperature and precipitation lapse-rate corrections as necessary). The GCM includes a dynamic vegetation module[91] rather than prescribe Pliocene-specific paleovegetation.

We also leverage the model matrix climatologies to provide time-evolving ocean temperatures to the ISM. We establish an empirical relationship between water temperatures of the upper mixed layer of the GCM ocean and subsurface (400 m water-depth) temperatures that influence grounding lines and are used to force the ice sheet model. This relationship was established from fully coupled atmosphere/ocean climate models spanning the last deglaciation[92], the last interglacial (lig127k CESM experiment)[93] as well as the warm Pliocene (PlioMIP2 CCSM4-Utr experiments under 400 and 560 ppm)[94]. CCSM4-Utr has relatively low climate sensitivity and good data-model agreement compared to the PRISM4 dataset[95], and uses the same model as the Liu et al.[92] deglacial simulation. CCSM4-Utr is one of the warmest PlioMIP2 ensemble members[95]; we use this simulation as a warm end-member to establish the ocean temperature scaling methodology. Our higher-$CO_2$ snapshot climatologies fall directly in the middle of the PlioMIP2 ensemble spread (Supplementary Fig. S6). Using these datasets, we relate mixed-layer temperatures in the 50–60°S latitude band with 400 m temperatures of the ocean grid cells nearest the ice sheet margin (Supplementary Fig. S7). A subsurface ocean temperature scaling can therefore be calculated for each GCM 50–60°S mixed-layer temperatures within the model matrix; as the model steps forward in time, the matrix weighting scheme determines a uniform anomaly correction to a modern ocean climatology[96] for each timestep.

A drawback of this ocean scaling approach is that it preserves the spatial structure of modern ocean temperatures throughout the model simulation, despite the transient and dynamic nature of ocean structure through time and potential feedbacks with ice growth[97]. We mitigate this issue by assuming that modern ocean temperatures are representative of warmer-than-modern times throughout the Pliocene (following the scaling approach as described above), but for colder-than-present times, the Liu et al.[92] glacial-state ocean is more representative. Therefore, when GCM matrix ocean temperatures fall below modern, the ice sheet model input ocean climatology is scaled between a modern ocean[96] and a glacial-state ocean at 20 ka[92], as shown in Supplementary Fig. S7. This approach avoids extrapolating the significant ocean warmth currently observed offshore the Amundsen/Bellingshausen region[98] throughout Pliocene glacial periods.

The matrix method relies on a high-resolution $CO_2$ time series in order to establish the appropriate climatology for each timestep. Although recent work has filled in many gaps in the Pliocene $CO_2$ record[99], studies that reconstruct $CO_2$ fluctuations at orbital-scale resolution remain limited in time (e.g., de la Vega et al. (MPWP)[88]; Chalk et al.[100], Hönisch et al. (MPT)[101]; Martinez-Boti et al. (Late Pliocene)[102]). Splicing together proxy records introduces possible $CO_2$ variability arising from differences between sites and between proxies, further complicating the use of a continuous proxy-based $CO_2$ record to drive the ice sheet model through time. We therefore employ an alternative method of reconstructing past $CO_2$ variability using a benthic $\delta^{13}C$-based proxy for atmospheric $CO_2$ (cf. Lisiecki[103]). Benthic $\delta^{13}C$ records in sediment cores reveal changes in deep water ventilation and deep ocean carbon storage[104], and therefore can be used to infer atmospheric $CO_2$; specifically, Lisiecki[103] found that a modified $\delta^{13}C$ gradient between the deep Pacific and intermediate North Atlantic ($\Delta\delta^{13}C_{P-NA/2}$) correlates well with $CO_2$ measured in ice cores across the last 800 ka. In this work, we extend the $\Delta\delta^{13}C_{P-NA/2}$ proxy further back in time to 4.5 Ma, producing a continuous orbital-scale $CO_2$ time series spanning the Pliocene.

The relationship between ocean $\delta^{13}C$ and atmospheric $CO_2$ becomes increasingly uncertain as we extend this proxy further back in time; changes in ocean circulation, carbon burial, and paleoproductivity can alter these $\delta^{13}C$ records, as well as long-term trends in weathering and tectonics that impact the global carbon cycle. Although these processes may have modified the absolute values of benthic $\delta^{13}C$ on long-term timescales, we assume that the timing of glacial/interglacial cyclicity preserved in these records remains robust. Therefore, after stacking the individual $\delta^{13}C$ records, we scale the resultant $\Delta\delta^{13}C_{P-NA/2}$ curve based on the mean and amplitude of boron isotope-based $CO_2$ reconstructions during the Pliocene[99] (Supplementary Fig. S8). The timing of ice sheet grounding line fluctuations is sensitive to this highly uncertain paleo-$CO_2$ formulation, though the amplitude is robust (model simulations across a time interval where orbital-scale reconstructions are available (3.3–2.6 Ma) produce similar amplitudes of glacial cyclicity as models forced by the $\Delta\delta^{13}C_{P-NA/2}$ $CO_2$ proxy).

Although our $\delta^{13}C$-derived $CO_2$ proxy varies widely, model ice sheet behavior does not directly mirror the $CO_2$ time series forcing (Fig. 3a, b). However, the simulated period of receded ice from ~3278 to 3142 ka was likely driven by elevated $CO_2$ in the proxy time series which may be an artifact of deep ocean reorganization rather than a change in deep ocean carbon, indicating elevated atmospheric $CO_2$.

## Geologic records and model-data comparison

Below we review the currently available geologic records of Pliocene ice sheet behavior, organized by region, along with a description of model-data agreement. We synthesize these records into specific data-based criteria for model evaluation (Supplementary Table S1), which we then use to assess each simulation in the model ensemble (Fig. 2). Model simulations are evaluated with respect to each criterion, producing a model-data agreement score ('1' for simulations deemed consistent with the geologic record, '0' for a poor fit or unclear model-data comparison, or '-1' for a model that violates the geologic record). Each criterion is given a confidence weighting that reflects the strength of the data interpretation or the robustness of the model-data comparison; given the wide range of data quality and ambiguity or certainty around proxies and data interpretation, the weighting factor correspondingly varies exponentially, from '10' (most confident) to '5' (less confident) to '1' (least confident). The model simulation score is calculated from the sum of the model-data agreement scores multiplied by the weighting factor for each criterion (Fig. 2).

The computational effort of performing an ensemble of multimillion-year simulations requires a 40 km model spatial resolution, so our interpretation of the geologic record and model-data comparison efforts are correspondingly coarse resolution. For example, we interpret modeled ice sheets that extend across *most* of the exposed continental shelf as being consistent with geologic records of grounded ice at or near the shelf break. We make allowances for these slight discrepancies because the position of the continental shelf edge was changing throughout the Pliocene in many regions, as the ice sheet actively eroded the bed, prograded the shelf, and constructed trough mouth fans.

We also note that our simulations do not produce a significant M2 glaciation (models simulate a glacial period at 3.32 Ma, but it is not

significantly stronger than other glacials). Our model is forced by time-evolving $CO_2$ and insolation; the $CO_2$ proxy dataset (Fig. 3b) does not indicate particularly low $CO_2$ at this time, and although $CO_2$ is thought to play a secondary role in triggering the M2[88], our model does not produce an orbitally driven glaciation either.

Amundsen Sea Embayment: Integrated seismic stratigraphy, drill core physical properties, and sedimentological data reveal dynamic WAIS behavior across the modern Amundsen Sea continental shelf during the early Pliocene (≥ 8 glacial advances out to the continental shelf break) and late Pliocene (≥3 glacial advances), interrupted by a period of prolonged mid-Pliocene ice sheet retreat from about 4.2–3.2 Ma, dubbed the Pliocene Amundsen Sea Warm Period (PAWP)[22,23]. Buried continental shelf grounding zone wedges identified from seismic records indicate that at least four glacial advances occurred during this million-year warm period, but were separated by long periods of ice sheet retreat spanning multiple glacial/interglacial cycles that buried these grounding zone wedges in hemipelagic sedimentation. This extended warm period is associated with high diatom contents and low terrigenous sedimentation rates in a sediment core spanning these seismic packages, suggesting reduced glacial processes on the Amundsen Sea continental shelf that trigger downslope transport of sediments from the shelf to the core site[22].

Most model results independently correlate with this interpretation, reproducing multiple dynamic WAIS advances across the continental shelf during the early and late Pliocene, with a long period of ice sheet retreat during the PAWP (Supplementary Fig. S9). End-member models with the highest sensitivity to ocean temperature or maximum MICI parameters are not able to grow sufficiently far across the continental shelf. Model-data comparison for this region is designated "most confident" given the detailed history of glacial expansion and retreat across the Pliocene.

Ross Sea Embayment: Large-scale unconformities along the Ross Sea continental shelf record periodic erosive ice sheet advances across the continental shelf during the Pliocene. Seismic stratigraphy mapping has identified at least 7–10 episodes of widespread glacial advance of the WAIS and EAIS into the Ross Sea during the Plio-Pleistocene[24–26], although the exact ages of these unconformities remain relatively unconstrained. At the ANDRILL-1B drill core in the western Ross Sea, periodic open ocean conditions alternated with grounded ice advance across this region throughout the Pliocene[12,27]. In all, 13 glacial erosional surfaces are identified during the later Pliocene (2.6–3.4 Ma), with no evidence for glacial advance before 3.4 Ma[12,27]. Paleoenvironmental reconstructions at this site reveal an extended warm interval from 4.5 to 3.4 Ma characterized by open ocean conditions, increased sea surface temperatures, and minimal marine-based ice and summer sea ice, followed by cooling and glacial expansion at about 3.3 Ma[12].

Most simulations produce frequent glacial expansions beyond the modern grounding line, but only models with low sensitivity to ocean temperatures advance all the way to the shelf break as indicated by the geologic record. The limited continental shelf expansions in most simulations may be due to model resolution issues or uncertainties in sub-ice-shelf bathymetry under the Ross Ice Shelf. In general, model ensemble members that produce glacial expansions across the shelf during the later Pliocene also advanced during the early Pliocene, despite the lack of geologic evidence for glacial expansion during that time. In the Ross Sea, the model ensemble generally reproduces an extended warm interval, but slightly offset from the geologic record (~4.1–3.2, rather than 4.5–3.4 Ma as in ref. 12). Model-data comparison for this region is designated as "most confident" given the detailed history of Pliocene glacial expansion and retreat.

Wilkes Subglacial Basin: Drill core data suggest that the ice sheet periodically advanced to the continental shelf break and then retreated inland hundreds of kilometers across the Wilkes Subglacial Basin during the Pliocene. On the continental shelf, alternating diamicts and open-marine sediments reveal dynamic glacial advance and retreat behavior[30,31]. Offshore turbidite deposits record periods of glacial advance to the continental shelf edge (≥12 advances from 4.5 to 2.6 Ma)[32,33]; ice sheet advance to the shelf edge and the onset of grounding line retreat is associated with pulses of iceberg-rafted debris[33,34]. During warm interglacial periods, diatom-rich/bearing muds accumulated offshore, with terrigenous components sourced from far inland suggesting large-scale grounding line retreat into the subglacial basin[32,44]. Large-scale glacial retreat across the Wilkes Subglacial Basin is also inferred from observations of iceberg-rafted debris pulses collected offshore Prydz Bay, which are attributed to destabilization and large-scale ice collapse in the Wilkes Subglacial Basin and Aurora Subglacial Basin under warmer-than-present conditions[39,40]. Recent work on marine sediment provenance offshore Wilkes Subglacial Basin provides a spatial constraint on 'large-scale' ice collapse events[45]: if the ice sheet margin periodically retreated by several hundred kilometers into the Wilkes Subglacial Basin, glacial erosion of the geochemically distinct Adelie craton region would be detected in offshore Pliocene sediments (as in warm Miocene intervals[105]). Erosion of significant amounts of Adelie Craton material in times of peak Pliocene warmth are, however, not observed in Wilkes Subglacial Basin provenance records[32,44], suggesting near total loss of the marine-based ice in the Wilkes Subglacial Basin did not occur.

Pliocene marine diatoms have been found in the Sirius Group formation, which outcrops along the Transantarctic Mountains. Here we do not use these data as an explicit model constraint because multiple interpretations explain the presence of these diatoms; they could indicate a shallow inland sea depositional environment following WAIS collapse[41], but more recently have been hypothesized as windblown grains from ice-free Wilkes or Aurora subglacial basins[42,43].

Most model simulations produce periodic glacial advance and retreat across the continental shelf, except for ensemble members with a high sensitivity to ocean warming which never advance all the way to the shelf break. Model members with MICI physics set to zero or low values ("HFoff", or "HFlow") do not retreat enough across this region to satisfy the geologic evidence for large-scale collapse[32,39,40,44]; however, models with maximum parameterized MICI sensitivity ('HFmax') produce frequent ice sheet collapse into the geologically contraindicated Adelie region[45] (also Shakun et al.[46], see "ANDRILL catchment region" below).

In addition, all models (barring "HFoff", which does not retreat sufficiently) show an episode of retreat at around 3.55 Ma, consistent with the pulse of Wilkes-sourced ice rafted debris reaching Prydz Bay[38,39].

Model-data comparison in this region is split into two categories: the compilation of numerous multi-proxy geologic evidence of grounding line advance and retreat; and the geochemical constraint on maximum extent of ice retreat, both designated a "most confident" model constraint.

ANDRILL catchment region: Cosmogenic nuclide concentrations in ANDRILL-1B sediments are extremely low[46]. These sediments are interpreted to reflect ice dynamics across the relatively large land catchment area delivering sediments to the AND-1B core site throughout the Pliocene; the low nuclide concentrations therefore preclude large-scale or periodic land exposure of much of the Transantarctic Mountain region or the southernmost part of the Wilkes Subglacial Basin.

This is consistent with model simulations, which never deglaciate the terrestrial Transantarctic Mountain region even during episodes of marine ice sheet collapse. The indirect nature of this evidence leads to a 'less confident' model-data comparison designation. This dataset provides a similar constraint to the geochemical evidence for Wilkes Subglacial Basin ice sheet retreat; model ensemble members where deglaciation extends across the southernmost Wilkes Subglacial Basin also erode into the Adelie craton region, which is precluded by sediment provenance analysis[45] as discussed above.

Aurora Subglacial Basin: The Aurora Subglacial Basin is substantially less studied than Wilkes Subglacial Basin. Large-scale glacial retreat across this region is also inferred from observations of far-traveled iceberg-rafted debris[39,40]. However, geophysical interpretations directly conflict, suggesting that grounding line retreat across the Aurora Subglacial Basin did not exceed ~150 km inland from its modern position since the Miocene[49] which precludes large-scale ice sheet collapse across this region during the Pliocene.

Only model members with zero (or low) parameterized sensitivity to MICI mechanisms prevent more than 150 km of grounding line retreat during past warm periods. These parameter combinations are inconsistent with the geologic record in other catchments, supporting the interpretation of Pliocene ice loss across the Aurora Subglacial Basin[39,40]. Since model-data comparison in this region is ambiguous, it is designated "least confident" due to the conflicting geologic reconstructions.

Prydz Bay: Periodic glacial advances eroded large-scale glacial unconformities across the Prydz Bay continental shelf during the Pliocene[35], constructing a large trough mouth fan and progradational continental slope deposits[36], and depositing ice-proximal diamictites across and beyond the continental shelf[37]. Inland outcrops of open-marine diatomaceous sediments in the Lambert Graben rift flank[47,48] suggest that these large-scale ice advances were punctuated by periods of ice sheet and ice shelf retreat by hundreds of kilometers. Although the ages of these retreat events are only loosely constrained, glacially reworked marine fossils in these outcropping formations date between 5.8 and 3.6 Ma at the Bardin Bluffs Formation[47], 4.9–4.1 Ma at the Larsemann Hills[106,107], and 4.2–4.1 Ma in the Vestfold Hills[47,108]. Offshore records similarly reveal cyclic ice sheet advance and retreat. Detailed analysis of iceberg-rafted debris accumulation rates indicates a long period of receded ice, both grounded ice and ice shelf, between 4.6 and 4.0 Ma[38]. After 3.3 Ma, IRD accumulation rates dramatically increase in amplitude and variability, with provenance data indicating more contribution from distal Wilkes Subglacial Basin sources[40]. This region is designated "most confident" due to the multi-proxy agreement of dynamic grounding line behavior.

Model ensemble members with zero, low, or intermediate MICI parameters do not produce grounding line retreat beyond the modern configuration, or ice sheet and ice shelf recession upstream of the glacially reworked open-marine sediment outcrops during warm interglacials, as inferred from the geologic record. None of the model members show reduced ice extents between 4.6 and 4.0 Ma; the modeled warm period occurs later (4.1–3.2 Ma). No modeled change in ice behavior is observable after 3.3 Ma; this may be when outer shelf erosion intensified and the trough mouth fan was constructed (3.9–3.6 Ma)[109,110], which likely influenced ice sheet dynamics across this time period but are not physical processes currently represented by the model.

Weddell Sea Embayment: Persistent sea ice conditions in the Weddell Sea have long hampered data collection efforts, so only limited information is available regarding ice sheet dynamics during the Pliocene. Seismic surveys of the eastern Antarctic Peninsula (northwestern Weddell Sea), correlated to the nearby SHALDRILL drill core, reveal 10 Pliocene ice sheet grounding events on the continental shelf[11], but it is unclear if these are related to WAIS fluctuations in the Weddell Sea. Seismic surveys of the Crary Trough Mouth Fan (on the other side of Weddell Sea, correlated to site 693 ODP leg 113), reveal extensive mass transport events during the Pliocene[28], associated with continental slope progradation as the ice sheet periodically advanced to the outer shelf edge[29].

All model simulations produce periodic glacial advance and retreat across the Weddell continental shelf; only extreme end members with high sensitivity to ocean temperatures and MICI parameters do not expand sufficiently. Model-data comparison in this region is 'least confident' given the lack of data coverage across the Weddell shelf, and the relatively unconstrained timing of trough mouth fan formation.

Interior nunataks: Cosmogenic nuclides measured in situ at exposed mountain peaks record episodes of interior ice sheet thickening during the Pliocene[50], attributed to increased precipitation during warm interglacials. Four locations reveal episodes of Pliocene ice thickening: Sør Rondane Mountains in Dronning Maud Land (+400 m between 2.5 and 3 Ma, possibly +600, before 4 Ma); Grove Mountains in Princess Elizabeth Land (+150–220 m at least once before 3.5 Ma); McMurdo Sound-Dry Valleys area in Victoria Land (+700 at least once before 2.8 Ma); and Shackleton range in Coats Land (+750 m at least once before 3 Ma).

All model members simulate ice sheet thickness changes of many hundreds of meters at each the specified nunatak locations; however, only models with lower parameterized sensitivity to ocean temperatures produce sufficient thickening (specifically, a difference arises at Dronning Maud Land and Victoria Land sites). This model-data comparison is designated 'least confident' for two main reasons: (a) the model grid resolution of 40 km does not resolve the mountain peaks where data were collected, and paleo-ice dynamics around nunataks are influenced by processes occurring at the sub-grid scale[112]; and (b) modeled interior ice thickening is heavily influenced by parameterizations that are not explored here, for example, relating to precipitation and accumulation.

Pirrit Hills nunatak: As an ice sheet grows and shrinks across multiple glacial cycles, the cyclic exposure of cosmogenic nuclides along a mountain peak can be measured to reconstruct 'fraction of time spent ice-covered' at each sampled elevation[113,114]. These measurements, along an elevation transect, can be directly compared to the frequency behavior of model ice sheet thickness fluctuations across millions of years[51]. The only location with sufficiently detailed cosmogenic nuclide data to construct an ice cover frequency curve is currently the Mt. Tidd Nunatak in the Pirrit Hills.

None of the model simulations produce a similar pattern of cyclic exposure and ice cover frequency as indicated by the transect of cosmogenic nuclide exposure ages at the Pirrit Hills. This model-data comparison is designated 'least confident', primarily given the coarse-resolution model grid size which complicates model-data comparisons, but also due to the fact that the Pirrit Hills ice thickness frequency dataset integrates glacial fluctuations across the last ~5 Myr while our modeled ice thickness frequency curves reflect ice sheet behavior during the Pliocene only.

## Data availability

Model output can be accessed via Zenodo (https://doi.org/10.5281/zenodo.12657538), including model fields of ice sheet thickness and velocity, bed topography, and calculations of ice sheet volume and global sea-level equivalents.

## Code availability

The ice sheet model code is from ref. 2. The specific version used here is available from the author upon request.

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

## Acknowledgements

The authors thank James Rae for discussions on developing our $CO_2$ forcing methodology, Sandra Passchier for guidance on integrating geologic data, and Michiel L.J. Baatsen for providing CCSM4-Utr model output. The authors acknowledge high-performance computing support from Cheyenne[115] provided by NCAR's Computational and Information Systems Laboratory, sponsored by the National Science Foundation. Support for ARWH and RMD was provided by NSF-OPP 2035080. E.G. is supported by a Royal Society fellowship and NERC award NE/T007397/2. J.M. is supported by NERC award NE/W000172/1.

## Author contributions

A.R.W.H., E.G., D.P., and R.M.D. conceived the project, performed simulations, and analyzed results. J.M. provided interpretations of the $\varepsilon_{Nd}$ dataset for model-data comparison. A.R.W.H. wrote the paper. E.G., D.P., J.M., and R.M.D. contributed feedback on the manuscript.

## Competing interests

The authors declare no competing interests.
