## [Peer Review File · Nature Communications]

Geologically constrained 2-million-year-long simulations of Antarctic Ice Sheet retreat and expansion through the PlioceneREVIEWER COMMENTS

Reviewer #1 (Remarks to the Author):

This paper represents a significant step forward in the use of this particular formulation of ice sheet dynamics. The evidence for this parameterisation of ice sheet retreat is lacking and it is an unnecessary addition to produce both modern and past observations (Edwards et al., 2019). Therefore, much more groundwork is required to test and understand its behaviour and this manuscript represents a good step in this direction. However, some significant clarifications are required in this manuscript, so that an audience not immersed in ice sheet and climate physics can understand it clearly.

The Pliocene is an important test-bed of ice sheet models, as it is the last period of similar to modern greenhouse gases, sustained increased global temperatures and the best-attested period of Antarctic ice sheet retreat. So, this a good choice of time period for this study. However, the matrix of climate models that are used within this study do not include a Pliocene climate state. Even with the ocean temperatures modified using a Last Glacial Maximum (LGM) simulation (albeit a rather old simulation) when they fall below modern temperatures, the same is not extended to warmer than modern temperatures, which could have produced a much more Pliocene like climate state. The CCSM4-Utr model's Pliocene simulation was used to relate surface and sub-surface temperatures, so not using it in the same way as the LGM simulation needs justifying.

It needs to be made clear in the main text of the paper exactly what climate forcings are going into these transient simulations. There are only 2 atmospheric carbon dioxide levels (285 and 421ppm), three orbital configurations (please also state these time periods) and three ice sheet configurations, none of which incorporate any East Antarctic Ice Sheet retreat. This means there is very limited ice sheet climate feedbacks in East Antarctica and using a slab ocean model means there is no ocean circulation changes in these simulations. The key evidence for the Pliocene being warmer than today is increases in sea surface temperature and the best explanation for this is changes in ocean circulation, brought about by geological changes in ocean gateways (Otto-Bliesner et al., 2016).

Given the above and the fact that the Charney sensitivity of this model is on the lowest end of modern estimates, I would expect the warming of the Pliocene climate in these simulations to be particularly low. A higher sensitivity climate model with ocean circulation feedbacks and set up with Pliocene feedbacks (palaeogeography, vegetation and more ice loss), could potentially see much more warming; close to double from non-CO2 and WAIS Pliocene feedbacks (Lunt et al., 2012) and climate sensitivity could be twice as much and still be within IPCC best estimates (IPCC AR6). As such, the modelled EAIS may not require additional physics to collapse and, indeed, in the PLISMIP experiments ice sheet models without these additions are able to simulate significant retreat in East Antarctica (Dolan et al., 2018).

Using the direct proximal evidence for the Pliocene Antarctic Ice Sheet seems like a good option for evaluating these simulations. However, the threshold between the simulations that are taken as geological constrained and those dismissed is extremely narrow. The two simulations considered as "best-fit" and "highest fidelity" are inconsistent with two of the 10 lines of evidence and a poor fit to two more and you use these to state that the simulations suggest that "the Antarctic Ice Sheet is highly sensitive to past and future warm conditions". However, one of your simulations (OC3-HFlow-area), with only 5 msle of AIS variability, is inconsistent with two lines of evidence and a poor fit to three. It seems to me that the available evidence is not of sufficient quality to rule out scenarios based on one extra unclear misfit to the data, especially with the magnitude of the claims that are produced from this. It needs to be made clear in the manuscript that the available data and these simulations really do not rule out a very stable Antarctic Ice Sheet, at least given the assessment made here.

There are a couple of significant changes that need to be implemented for this manuscript to be suitable for publication, namely:

- The close match between the geological record and simulations with very little ice sheet fluctuation needs to be recognised much more in the manuscript. Particularly, this needs to be included in the summary statement and the abstract. I would recommend finishing the final sentence of the summary with the clause "without ruling out very stable ice sheet parameter sets". In the abstract, I would rewrite the sentence starting on line 11 to be "Geologically consistent

simulations reveal a dynamic ice sheet that could have contributed up to 25m of sea level change across glacial-interglacial cycles, although very stable ice sheet configurations cannot be ruled out.”

- The climate model simulations that make up the matrix driving the ice sheet model needs to be included in the main text, somewhere around line 48. This description should include the CO₂ levels, orbital configurations (including times represented) and Antarctic Ice Sheets incorporated (maybe with an image of the latter). It should also mention the things that aren't included, especially Pliocene palaeogeographic changes, climate feedbacks from EAIS loss and changes in ocean circulation.

There are also a number of changes that would improve the manuscript:

- The opening sentence of the abstract makes it sound like the Pliocene is rarely as warm as modern, whereas temperature records suggest that it was warmer for very long periods of time, at least 100s of thousands of years, if not millions of years. Please rephrase to reflect this.
- Line 5 of the abstract states that you are using coupled ice sheet and climate models. This is only true in the most generalised of senses and not in the normal scientific meaning of this phrase. The following would be a much clearer sentence to include here. “Here we employ an ice sheet model with forcing provided by a matrix of climate model simulations to simulate transient Antarctic ice sheet cyclicity from 4.5-2.6 Ma. This provides ...”
- Line 8: From this manuscript I cannot see how the data is used to “constrain the model” or “ensure geologic compatibility”. Better verbs would be to “evaluate” and “test” respectively.
- Line 10: I do not see any point in the manuscript where the Antarctic geological record is extrapolated. Either remove this sentence or make clear what this extrapolation is.
- Line 26: Do you mean the Pliocene sea level signal? It currently reads as the future sea level, since this is the last sea level record you mention, which would make no sense.
- Line 31: using a constant climate could melt the ice sheet rather than build it up, just depending where in a climate cycle is simulated.
- Line 32: Many modelling studies (e.g. PLISMIP) incorporate the uncertainty due to hysteresis (and it is relatively straightforward to do so). Maybe add “although modelling can take this into account” at the end of the sentence.
- A better description of how subsurface temperatures are calculated should be included in line 49.
- Line 139: I think it is important to point out here that although your model cannot retreat into the Wilkes Subglacial Basin without MICI that other models are able to reproduce this, albeit with different climate forcings (e.g. PLISMIP)
- Section 2.3: Please could you explain why warming episodes in your model setup do not increase the height of the Antarctic Ice Sheet, through increased precipitation. Is this something the model is unable to do or does not reproduce? This relationship is attested by present day observations, future ice sheet simulations and Pliocene climate and ice sheet modelling, so why does this not hold for your modelling setup?
- Line 196: It is fair to say that there are significant sources of uncertainty in oxygen isotope based sea level estimates, but I am a little uncomfortable dismissing them as simply unreliable. Maybe something like “though $\delta^{18}O$ estimates have significant uncertainties, particularly when the multiple factors that can affect the signal are considered¹⁶.”
- Line 306: I am confused about your comments on the mPWP, given that the results presented here show the highest CO₂ concentration (and therefore presumably temperatures) and the longest period of retreated Antarctic Ice Sheets during the mid-Pliocene warm period. It is very plausible that the early Pliocene was warmer than the mid-Pliocene, which is suggested by long climate records, and especially so in the Southern Hemisphere when the Isthmus of Panama was open (although you wouldn't capture this here).
- Line 322: I don't know of anyone who suggested that the mPWP was the warmest period of the Pliocene, globally or in Antarctica. So please give a reference here, if you are trying to refute something in particular. In fact many people have suggested the early Pliocene was significantly warmer than the mid-Pliocene (e.g. ANDRILL or Ohneiser et al., 2020).
- Line 378: I don't think it makes any difference, but it is worth noting that CCSM4-Utr is one of the warmer models in PlioMIP2 and by some estimates has the strongest response to non-CO₂ Pliocene boundary conditions (Haywood et al., 2020).
- Line 385: It should be noted that modern melting of Antarctica is driven by upwelling of deeper waters penetrating into glacial troughs on the Antarctic shelf (Walker et al., 2007; “Oceanic heat transport onto the Amundsen Sea shelf through a submarine glacial trough”). So I don't think this formulation would do a good job of simulating observed melting of Antarctica, as it is not closely related to the mixed layer temperatures. Is this something you have considered?
- Line 393: You have data from the CCSM4-Utr Pliocene model, so please justify why the same

scaling was not applied when temperatures were above modern.

- Line 559: Please note here that there are marine sediments above the modern ice surface, which suggests significant topographic change that is not included in these models.
- Line 599: should say "model-data" rather than "mode-data".
- Table 1: "Sea level amplitude" is a good metric, but most attempts to produce a Pliocene sea level estimate aim at maximum sea level. It would be great to have another column with the maximum sea level in metres above present day sea level, so the simulations can be compared to data.
- Figure M3 is an absolutely key figure, in that it shows the forcings that go into the model, so needs to be in the main paper. It could perhaps be added as panels in Figure 2. It would also be great to have the three orbital forcing configurations that are actually simulated added to panel c with a symbol.

Reviewer #2 (Remarks to the Author):

Review Halberstadt et al.

GEOLOGICALLY CONSTRAINED ANTARCTIC ICE SHEET SIMULATIONS: 2 MILLION YEARS OF GLACIAL RETREAT AND EXPANSION THROUGH THE WARM PLIOCENE

Halberstadt et al. present an ensemble of paleo ice sheet simulations spanning the time from 4.5 – 2.6 Ma BP. Employing a matrix/climate index offline forcing approach and a coarse resolution model setup (40 km) they assess the evolution of the AIS during glacial-interglacial cycles and find extended periods of retreated ice sheet configuration driven by MIS1 and MIC1 but also individual periods in which the AIS attains a full glacial state. They calibrate and discuss their model results in light of the available proxy data finding good agreement with i.e. marine sediment cores suggesting sustained ice free conditions in different ocean sectors. They conclude that an intermediate parameterization of MIC1 in PSU-ISM yields the best results with respect to the proxy records, showing that low MIC1 parameters do not produce sufficient ice retreat compared to proxies and high MIC1 does not allow for an expanded ice sheet margin. Both cases are not in line with current estimates from paleo-proxy reconstructions. This is an important step in calibrating a highly uncertain process such as MIC1 in an ice sheet model. The manuscript is very well written and easy to follow despite the complexity of the approach and the various proxy records taken into account. The method section is substantial and very well written shedding light on the more involved aspects of the model setup employed. The main flaw of the study is the model resolution (40 km) which is very coarse for current standards. While I realize that simulations covering such a long time-span can be prohibitive from a computational standpoint it should be possible to at least produce a higher resolution run of the best fit ensemble-members highlighted in table 1. In his highly cited paper from 2009 Pollard and DeConto (both co-authors here) carried out 5 Ma of model simulations on 40 km including nested 10 km res simulations as well. I am aware that model physics were probably not as complex as the current version of PSU-ISM but say 15-20 km res simulations for a subset (see above) should be manageable and would improve both the quality of the paper as well as the robustness of the results. The second aspect I was missing is a PD or PI calibration of the model setup. As the authors correctly state, proxy recs for the Pliocene are sparse and sometimes of qualitative nature while we have a very good coverage of the current AIS state. I am aware that it is almost customary to reference previous instalments of the model stating that calibration has been carried out there and the model performs acceptably under well constrained reference conditions. It would be nice to see some more information on that in the supplements i.e. equilibrium thickness change and grounding zone positions for e.g. PI-cntr conditions.

This being said I congratulate the authors on an interesting study and I would suggest publication in light of my points above and some minor comments outlined below.

Best Regards,

Johannes Sutter

L70 .. Best-fit model simulations are used to extrapolate pinpoint geologic records, disparate in space and time, into a continuous and geologically constrained reconstruction of AIS contribution to Pliocene sea level.

This is a nice idea, what about uncertainty assessment due to ISM and boundary conditions?

L93 "trough mouth fan development"

maybe provide a reference as this manuscript is intended to be understandable by a wider (non-expert) audience.

L188 "GMSL ranges from +18m during an Antarctic interglacial/Greenland glacial period (+18m from the Antarctic Ice Sheet plus 0m from an approximately modern-size Greenland) to -2m during an Antarctic glacial/Greenland interglacial (-7m from Antarctica plus +5m from Greenland), resulting in GMSL amplitude of 20m."

I don't quite understand the reasoning here. During a period of NH glacial advance (I'm not sure whether the term Greenland glacial period is fortunate here as globally you're either in an interglacial or glacial but not both at the same time) antiphased to SH retreat you can't exclude a Laurentide or Fennoscandian component growing so the assumption +0m is a bit unfounded. Likewise, why capping the max Greenland contr. to 5 m, PD GIS has about 7m SLE.

L211 "Specifically, zero or low MICI parameter values cannot generate enough grounding-line retreat across Wilkes Subglacial Basin (to produce large pulses of far-traveled iceberg rafted debris) or Prydz Bay (to deposit inland open marine sediments),"

Couldn't an expanded Wilkes Subglacial Basin ice cover (grounding line adv. + ice thickening) which quickly retreats to PD conditions also produce a large far-traveled iceberg armada signal). At least this would be an alternative explanation which might be as conceivable as a large inland retreat of the grounding zone.

L219 If MICI is a process which has driven ice sheet instability in the past, a valid parameterization for these past climate states (both on the warmer and colder spectrum than PD) should also be valid for projections such as done e.g. in DeConto et al. 2016. Otherwise, one would have to change parameterisations of MICI depending on the climate state which would not be very helpful I guess. Wouldn't it be possible to at least inform the parameter range which DeConto et al. used and therefore reduce the associated uncertainty in hindsight?

L226 "Ice sheet fluctuations are driven by variation in climate forcing" as well as internal dynamics/feedbacks?

L285 "In other words, our simulations indicate that the ice sheet thickening at coastal locations reported in Yamane et al. 2015 occurred during Pliocene glacials, while thickening at interior locations occurred during Pliocene interglacials. During these interglacial warm periods, however, ice sheet contribution to GMSL during warm periods was overwhelmingly dominated by mass loss at marine margins despite increased surface accumulation."

I guess this is somewhat to be expected as it basically corresponds to late Quaternary glacial interglacial thickening/thinning patterns in the interior EAIS where both ice core proxies as well as modelling suggest thickening during interglacials and thinning during glacials due to SMB changes.

L322 "Antarctica but earlier than the MPWP, suggesting that the globally recognized MPWP may not have been the warmest Pliocene period in Antarctica"

Did you test your model setup on the MPWP, would be interesting to see your model-response in comparison.

L324/325 You mentioned before that it is not straightforward to assess the DeConto et al 2016 parameterization with your approach here, but it would be still nice to contextualize this at least approximately. E.g. the MICI parameterization is on the lower/higher/mid range compared to what's been used in the 2016 paper.

Methods:

L336 I suggest to specify this a bit more, acceptably is a very subjective term in this context. To see caveats it would be good to know whether e.g. grounding line reversibility etc. is comparable

to the mean model response or belongs to the better/weaker performing models.

L340 what parameterization is used for sub-ice basal melt rates and what is the reasoning for the scaling parameters? I am aware that this is probably mentioned somewhere in ref 2 but a short sentence here would be helpful.

L341 maybe I'm reading this wrong, but does that mean, ice cliffs retreat 100 km, 200 km, 300 km or 1200 km per year? I'm sure this is not the case but for readers which are not familiar with the MICI parameterization in PSU-ISM it would be nice to quickly specify what this means in real wastage rates and under which conditions these are switched on.

L349 It would be great to see some delta plots (e.g. ΔH) for PD/PI conditions and e.g. ice extent for LGM in the supplements. This gives the reader a clearer idea about potential biases in the modelling approach.

L352/353 it would be good to have some sense of resolution-sensitivity. I suggest to run a shorter timeframe (couple of glacial-interglacial cycles e.g. 200 kyr) on e.g. 15-20 km to see how the model response will vary. This should be possible computationally for a subset of parameterizations (e.g. min/max and middle of the road). Alternatively a nested approach as in Pollard & DeConto 2009 could be reasonable.

L354 This means there's no model spinup? Probably not a larger issue due to multi-million year period you are investigating but would be good to mention how the initialization shock looks like.

L423-426 the geologic constrained compiled by you for this study could be a very valuable resource for the community in general if available from a public repository. If all the data is published somewhere and can be uploaded this would be a great bonus outcome of this study.

L427-430 I think a similar sentence should appear in the main part of the manuscript so the reader can interpret the results with all biases taken into account.

L508 How well can ice retreat be quantified from these sediment records as opposed to just saying there was retreat but we don't really know the baseline state of the ice sheet neither how much grl retreated. What I am getting at is whether these records are robust enough to make a statement about MICI parameterization. I assume there is grl retreat into the Wilkes Subglacial Basin and Aurora basin also without MICI, what is the reasoning behind that this is too small of a change (maybe I have missed this in the main manuscript).

L522 It is impossible to adequately resolve the transatlantic mountain range at the resolution considered here. There will always be too much ice accumulating as small outlet glaciers are insufficiently resolved and ice drainage is too slow. This is a typical problem with coarser resolution continental scale simulations so I'd be very careful interpreting results from this region.

Table 1 it would be interesting to disentangle in the matrix which element has low confidence/poor fit due to known model biases. As you mention in the manuscript, comparing ice thickness changes at Nunataks at this resolution is not very informative, and I'm wondering why one would include it at all in this assessment.

Figure 1, 2, 5: the 2d AIS plots are cut off at a latitude which masks the Peninsula and parts of the EAIS. I suggest to make the cut at a lower latitude.

Figure 1 and 2: from the time series it seems that some glacial-interglacial-glacial transitions don't show hysteresis (i.e. fast retreat, slow regrowth). It would be nice to hear the authors view on why this is (forcing, model parameterization, combination of the two)?

Figure 3: this is a nice way of visualizing the results. Maybe add cm/yr to the colorbar label.

Figure 4: while this is a nice way of illustrating the results I think one could move it to the supplements. It is clear from a physical point of view that deglaciation rates in cases of ice sheet collapse are much faster than growth rates (the latter being mainly fueled by a cessation of high surface and basal melt rates and not by nonlinear effects such as MISI and MICI). Seeing this quantified in a figure is nice but does not really show anything new per se.

Figure 5 I'm a little confused by the figure caption (d) your model results show interior EAIS thickening in interglacials and in the manuscript you attribute this to higher precip rates which makes sense and is what we see in late quaternary interglacials. However, in the caption of (d) you state, that your model produces higher interior precip rates for cold glacial orbits (keeping CO₂ at the same rate). Is this representative of your glacial states in the matrix method or just a special case? If it is representative than I don't understand the thickening in interglacials even though integrated SMB would be smaller. Please clarify and modify the respective paragraphs if needed.

Figure 6 I realize that this is supposed to be a qualitative figure, but it is difficult to disentangle what it illustrates. To begin with, the continental shelf edge is relatively close or far away for PD reference conditions. Is the vertical axis calibrated with respect to this? Is distance measured along latitude or nearest point? Does the lower limit of the y-axis represent the continental shelf edge and the upper limit is positioned according to scale? If so the lower end of the y-axis would be zero right? It would help to add y-axis ticks as well. From an aesthetic point of view, inlet (c) could be improved.

Reviewer #3 (Remarks to the Author):

Halberstadt et al. present model experiments to simulate the transient behaviour of the Antarctic ice sheet through the Pliocene (4.5-2.6 Ma). They also present a compilation of geological evidence to document advances and retreats during this time period. These geological evidence are used to constrain their model results. These simulations are clearly interesting are they intrinsically represent the transient nature of this time period while the majority of previous modelling works have focused on selected boundary conditions supposedly representative of typical "Pliocene" climate (e.g. warm/cold orbit, high/low co₂, etc.). I generally enjoy reading this paper as I think that the chosen methodology is sound and the results provide a better understanding of the Antarctic ice sheet dynamics through the Pliocene. However, I have a few comments that might deserved some discussions.

Major comments

1. Uncertainties and model constraints. I appreciate the authors' effort to compile available data to constrain their model results. That being said, I found the "constraints" offered by this data relatively weak as they don't offer precise indication on space nor time. Since these constraints are so weak I feel that many model experiments could in principle agree with them. The authors here have explored only a very restricted model parameter ensemble (sensitivity to oceanic temperature change, MICI and a methodological parameter). I feel that the ice sheet dynamics shown here is affected by some modelling choices: the ice sheet model used, the climate forcing used to generate the climate matrix, the parameter in the regional climate model used,... On top of that the greenhouse gas concentration reconstructions is far from being perfect and is associated with a considerable uncertainty. I do not expect the authors to quantify explicitly all these uncertainties as the simulations presented here are probably numerically expensive. However, after reading the paper, I have the feeling that one could have simulate different sea level amplitudes and different waxing/waning frequency using a slightly different setup. Perhaps at least a discussion on model biases and how these biases could affect the results could be added in the manuscript? And the same for the CO₂ curve?

2. Sensitivity to atmospheric and oceanic climate change. From the results presented it seems that without the MICI (HF off) the model is unable to simulate grounding line retreat in East Antarctica. From Fig. 1c it also seems that no ice volume lower than PI is simulated in HF off. For the forcings representative of the Pliocene we can expect some ice sheet retreat in East Antarctica without invoking the MICI. For example in recent intercomparison exercise for future warming scenario in ISMIP6 (Seroussi et al., 2020) or for the buttressing intercomparison exercise (Sun et al., 2020), some models produce East Antarctica grounding line retreat. The model used here might present a relatively weak sensitivity to the oceanic forcing. Also, I was wondering about the atmospheric sensitivity. There is no information on surface mass balance computation in the manuscript (PDD? lapserate?...), only precipitation changes are discussed. With the forcings used, do you simulate

surface melt? Does it explain part of the grounding line retreat simulated?

3. Early Pliocene ice sheet volume. I was surprised to see a systematically larger ice sheet than today for the earlier part of the record (Fig. 1 and Fig. 2). Is this linked with the CO₂ reconstruction based on $\delta^{13}C$? For this part of the simulation, we can expect a role of Northern Hemisphere ice sheets as the climate seems favourable to their growth. I do not think that Northern Hemisphere ice sheets for this time period are supported by geological evidence... More generally in the paper it could have been useful to discuss the fact that to sustain a large scale Antarctic glaciation it might require a cool climate that can allow for Northern Hemisphere ice growth.

4. Erosion and bedrock evolution. It is hard to imagine that the bedrock has remained the same as today over these long timescales. At present there are some deeply incised troughs in some areas that might be less marked during the early Pliocene for example. The amount of subglacial sediment may have been different as well. Is the bedrock used the one of the pre-industrial? The bedrock topography is a key driver for ice sheet dynamics and as such these considerations could have been more thoroughly discussed. In the model it would have been possible to use alternative and/or evolving bedrocks to partially quantify this effect?

Minor comments / questions

L. 370 Which topography the regional climate model use? And how the simulated climate by the RCM is downscaled to the ice sheet model? Using a lapse rate? Is there any precipitation correction?

Fig. M1 The GCM matrix lines for ΔT_{mixed} layer are all below the CCSM4 runs. If I understand correctly it means that the CCSM4 runs are all warmer than the GENESIS runs for this particular variable. Does a direct scaling really make sense then?

L. 437-441 There is a potential role of Northern Hemisphere ice sheets here that is disregarded in this paper.

Technical corrections

L.45 and L.335 The Schoof (2007) formulation of the flux at the grounding line is not really a parametrization.

L. 341-342 The value given for V_{CLIFF} are in m/yr right?

L. 376 "glacial cycle" -> deglaciation

Fig 1. What ESL means here? Volume above flotation in time with respect to the volume above flotation for PI?

Fig. 1 shows poor quality.

Reference

Schoof, C. (2007), Ice sheet grounding line dynamics: Steady states, stability, and hysteresis, *J. Geophys. Res.*, 112, F03S28, doi:10.1029/2006JF000664.

Seroussi, H. et al.: ISMIP6 Antarctica: a multi-model ensemble of the Antarctic ice sheet evolution over the 21st century, *The Cryosphere*, 14, 3033–3070, <https://doi.org/10.5194/tc-14-3033-2020>, 2020.

Sun S, Pattyn F, Simon EG, et al. Antarctic ice sheet response to sudden and sustained ice-shelf collapse (ABUMIP). *Journal of Glaciology*. 2020;66(260):891-904. doi:10.1017/jog.2020.67

**Response to reviewer comments on:
“Geologically constrained Antarctic Ice Sheet simulations: 2 million years of
glacial retreat and expansion through the warm Pliocene”
by A. R. Halberstadt et al.**

We would like to thank the reviewers for their constructive and insightful suggestions which have helped us to improve this manuscript. Reviewer comments are shown in black font, and the author response is shown in red italics. Author’s comments reference line numbers in the revised manuscript with Track Changes turned on (we also submit a revised version with the changes fully incorporated, as well as updated reference formatting).

Reviewer #1 (Remarks to the Author):

This paper represents a significant step forward in the use of this particular formulation of ice sheet dynamics. The evidence for this parameterisation of ice sheet retreat is lacking and it is an unnecessary addition to produce both modern and past observations (Edwards et al., 2019). Therefore, much more groundwork is required to test and understand its behaviour and this manuscript represents a good step in this direction. However, some significant clarifications are required in this manuscript, so that an audience not immersed in ice sheet and climate physics can understand it clearly.

We thank Reviewer 1 for their thoughtful consideration and constructive suggestions for making our work more widely accessible.

The Pliocene is an important test-bed of ice sheet models, as it is the last period of similar to modern greenhouse gases, sustained increased global temperatures and the best-attested period of Antarctic ice sheet retreat. So, this a good choice of time period for this study. However, the matrix of climate models that are used within this study do not include a Pliocene climate state. Even with the ocean temperatures modified using a Last Glacial Maximum (LGM) simulation (albeit a rather old simulation) when they fall below modern temperatures, the same is not extended to warmer than modern temperatures, which could have produced a much more Pliocene like climate state. The CCSM4-Utr model’s Pliocene simulation was used to relate surface and sub-surface temperatures, so not using it in the same way as the LGM simulation needs justifying.

This paragraph raises two concerns, which we address below in turn: first, that our matrix of climate models do not include a Pliocene climate state; and second, that we use a different ocean temperature scaling methodology for cooler-than-present times versus warmer-than-present times.

(1) We interpret the reviewer’s statement that our matrix of climate models “do not include a Pliocene climate state” to refer to the observation from Fig. M1 (now Fig S7) that the PLIOMIP CCSM4 ensemble members are significantly warmer than our climate model snapshots (colored vertical lines).

These CCSM4 simulations are, however, some of the warmest model members from the PLIOMIP ensemble. In the figure below, we compare surface air and ocean temperatures (SAT and SST) from our climate model snapshots to the full PLIOMIP ensemble (reproduced from Haywood et al., 2020, Fig. 7). Here we plot global temperature averages from 60N-60S (because south of 60S, our climate models incorporate either a glacial-size, modern, or vastly reduced ice sheet configuration; and these >60S surface temperature differences impact globally averaged temperatures). Here we show that our climate model snapshots span the bulk of the wide-ranging PLIOMIP SST/SAT space. Basically, by changing insolation and ice sheet topography, our warmer/higher-CO₂ Pliocene matrix snapshots (orange) span much of the climate variability represented in the PLIOMIP ensemble of models, although we do not simulate the extreme warmth represented in the warmest three PLIOMIP model members including CCSM4-Utr. (Note that our cooler/lower-CO₂ Pliocene snapshots (blue) expand below the PLIOMIP range of climates; the prescribed climatic inputs for PLIOMIP simulations were designed to reflect mid-Pliocene Warm Period conditions, rather than a range of glacial/interglacial conditions as we explore here).

Fig. S6. Climate model snapshots compared to PlioMIP2 ensemble members. Climate model snapshots using the GENESIS GCM with 421 ppm CO₂ produce Pliocene climatologies (surface air temperatures and sea surface temperatures) that fall within the PlioMIP2 ensemble range of variability.

Climate model snapshots with lower CO₂ expand below the PlioMIP2 range; PlioMIP2 simulations were conducted under 400 ppm CO₂ concentrations to reconstruct the warm mid-Pliocene, while we explore a wider range of glacial/interglacial conditions.

So if this model is one of the warmest in the PLIOMIP ensemble, why did we choose to use PLIOMIP CCSM4-Utr model output to establish our SST/subsurface ocean temperature scaling relationship? We made this choice because we use another CCSM4 simulation (Trace21k; Liu et al., 2009) to anchor the cooler end of this SAT/SST space in our ocean scaling relationship, and using products from the same modeling tool reduces some of the uncertainty associated with different model physics.

We have edited the manuscript to better reflect these considerations; we now include the above Figure S6, and add on L428-431: “CCSM4-Utr is one of the warmest PlioMIP2 ensemble members (Haywood et al., 2020); we use this simulation as a warm end-member to establish the ocean temperature scaling methodology. Our higher-CO₂ snapshot climatologies fall directly in the middle of the PlioMIP2 ensemble spread (Fig. S6).”

(2) We use the ocean scaling relationship in Fig. M1 (now Fig. S7) to apply a uniform temperature anomaly to a modern ocean climatology – but only for warmer-than-present time periods. For cooler-than-present time periods (e.g., Pliocene glacials), we scale between a modern ocean and a modeled glacial ocean.

Applying the same approach for warmer time periods (e.g., scaling between a modern ocean and a CCSM4 Pliocene ocean) would indeed produce much warmer oceans and in theory a stronger ocean forcing. Based on the large spread in Pliocene climates within the PlioMIP2 ensemble highlighted above, though, choosing a CCSM4-Utr Pliocene ocean as a warmest end-member may not robustly reflect a more “Pliocene-like” climate state; a PlioMIP2 multi-model-mean ocean product is arguably the best representation of a “Pliocene-like” climate state, but this MMM product does not include subsurface temperatures that the ice sheet model requires as input.

We agree wholeheartedly with Reviewer 1 that applying a different approach for warmer versus cooler time periods is methodologically messy and unsatisfying, though we still believe that it is the best choice for this work given the complexities and limitations in available climate model data discussed here. We outline our logic behind this decision, both here as a response to reviewer concerns, and also in the manuscript to provide necessary information to readers.

- In our original model setup, during the initial stages of this work, we applied the same methodology to derive ocean input for both cooler and warmer times (using the uniform temperature adjustment for all timesteps). A uniform temperature

scaling approach applied on top of a modern ocean climatology avoids making assumptions about the spatial and vertical heterogeneity of Pliocene ocean temperatures. Given the uncertainty in model products, and the grid resolution limitations of PlioMIP2 coupled global climate models, we feel that the modern structural distribution of ocean temperatures is currently the most appropriate assumption to make about the Pliocene ocean.

- *However, we noticed that none of our ice sheet model ensemble members ever grew ice across the Amundsen/Bellinghausen continental shelf. Ice advance in these models is precluded by a zone of enhanced warmth in the Amundsen & Bellinghausen seas that is a salient feature of the modern ocean climatology (World Ocean Atlas; Levitus, 2012; also described by Schmidtko et al 2014). Even with a negative uniform temperature anomaly (e.g., during Pliocene glacial times), a model ice sheet that advances to the Ross and Weddell continental shelf edge still cannot grow over the Amundsen Sea region due to these warm ocean temperatures. Given the geologic evidence that ice did advance across this area in the Pliocene, we concluded that the modern spatial occurrence of warmer ocean temperatures in this region was likely not present during Pliocene glacials, and we therefore implemented a different glacial-ocean scaling adjustment for cooler-than-present times that modified the spatial structure of ocean temperatures.*
- *Although we opted to implement a glacial ocean weighting approach for cooler-than-present times, we preserve the warmer-than-present uniform-temperature scaling because (a) we have no evidence for if the modern spatial temperature distribution is inappropriate for warmer times; and (b) the AIS is mostly terrestrial during the warmest times, where this scaling would be most impactful, and therefore the ocean warmth forcing isn't expect to significantly impact our results.*
- *Previous Pliocene ice sheet modeling work has applied a uniform temperature correction to a modern ocean climatology (e.g., DeConto & Pollard, 2016; DeConto et al., 2021), but no previous work has been able to simulate the transient evolution of the ice sheet (without inverting from the oxygen isotope record) and so we have no precedent for an ocean temperature input scheme that accounts for changing spatial structure of the oceans across glacial/interglacial cycles. This methodological dilemma highlights the difficulties arising from incomplete proxy based understanding of ocean dynamics in the deep past.*

Edits to L437-446: "A drawback of this ocean scaling approach is that it preserves the spatial structure of modern ocean temperatures throughout the model simulation, despite the transient and dynamic nature of ocean structure through time and potential feedbacks with ice growth (Hill et al., 2017). We mitigate this issue by assuming that modern ocean temperatures are representative of warmer-than-modern times

throughout the Pliocene (following the scaling approach as described above), but for colder-than-present times, the Liu et al. glacial-state ocean is more representative... This approach avoids extrapolating the significant ocean warmth currently observed offshore the Amundsen/Bellingshausen region (Schmidtke et al., 2014) throughout Pliocene glacial periods.

It needs to be made clear in the main text of the paper exactly what climate forcings are going into these transient simulations. There are only 2 atmospheric carbon dioxide levels (285 and 421ppm), three orbital configurations (please also state these time periods) and three ice sheet configurations, none of which incorporate any East Antarctic Ice Sheet retreat. This means there is very limited ice sheet climate feedbacks in East Antarctica and using a slab ocean model means there is no ocean circulation changes in these simulations. The key evidence for the Pliocene being warmer than today is increases in sea surface temperature and the best explanation for this is changes in ocean circulation, brought about by geological changes in ocean gateways (Otto-Bliesner et al., 2016).

Our smallest-ice topography is characterized by a collapsed West Antarctic Ice Sheet as well as significant East Antarctic Ice Sheet retreat; this has been added to the text, and we have also added a Figure SB showing climate forcings (CO₂, insolation, and matrix topographies). We believe that this clarification addresses the reviewer concerns above (i.e., we are indeed representing ice sheet/climate feedbacks in East Antarctica, although this was not clearly stated before). We now provide the details of the three orbital configurations (Fig. S5), and have added the time periods corresponding to these max/min/median configurations in the text (L52 and L409).

The slab ocean model does not produce full-depth ocean circulation (using a coupled climate-ocean-ice-sheet model is currently computationally infeasible for so many equilibrated matrix snapshots or continuously for millions of years). Given these limitations we think our approach is a reasonable compromise and stress that the uncertainties introduced by using a slab ocean are well within the range of uncertainties associated with Pliocene proxy reconstructions of ocean and air temperatures.

Figure S5: End-member parameters for the climate matrix snapshots. (a) CO₂ end-member values were chosen based on the maximum and minimum CO₂ concentrations reconstructed from an orbital-resolution dataset (de la Vega et al., 2020; across the KM5c interval). (b) Time periods characterized by maximum, minimum, and median Antarctic summer insolation values during the G17 interval provided obliquity, eccentricity, and precession values to construct ‘maximum-insolation orbit’, ‘minimum-insolation orbit’, and ‘median-insolation orbit’ matrix snapshots (respective values - obliquity: 23.80°, 22.71°, 22.89°; eccentricity: 0.0396, 0.0382, 0.0073; precession angle: 68.6°, 245.5°, 155.0°). (c) Surface topography is either ‘no marine ice’, with ice loss from marine-based portions of West and parts of East Antarctica (DeConto et al., 2021 Pliocene simulation); modern (Bedmap2); or a Pliocene glacial configuration from a simulation forced by a climate model output with 285 ppm CO₂, a ‘cold’ astronomical orbit, and eustatic sea level at -60m.

Given the above and the fact that the Charney sensitivity of this model is on the lowest end of modern estimates, I would expect the warming of the Pliocene climate in these simulations to be particularly low. A higher sensitivity climate model with ocean circulation feedbacks and set up with Pliocene feedbacks (palaeogeography, vegetation and more ice loss), could potentially see much more warming; close to double from non-CO₂ and WAIS Pliocene feedbacks (Lunt et al., 2012) and climate sensitivity could be twice as much and still be within IPCC best estimates (IPCC AR6). As such, the modelled EAIS may not require additional physics to collapse and, indeed, in the PLISMIP experiments ice sheet models without these additions are able to simulate significant retreat in East Antarctica (Dolan et al., 2018).

We address the two concerns highlighted in the above paragraph separately. First, that our Pliocene climate simulations would have a low amount of warming – while the climate sensitivity of our model is on the low end of the PlioMIP2 spectrum, it is within

the range of PlioMIP2 ensemble members, and is similar to CCSM4-Utr (see figure below). Fig. S6 (above) demonstrates that our climate model snapshots fall within the PlioMIP2 range of simulations (although we agree that with a higher climate sensitivity, our modeled climates would certainly be much warmer). We also note that there is significant uncertainty associated with geologic proxy reconstructions of paleo-CO₂, in addition to the uncertainty around the 'real' Earth system climate sensitivity in the Pliocene (e.g., the temperature response to CO₂ doubling); this large uncertainty is reflected by the large range of climate sensitivity values within the PlioMIP2 ensemble.

With vastly more resources and unlimited time, we would perform a more comprehensive model ensemble and include different climate models with different climate sensitivities, impose reconstructed Pliocene paleovegetation, vary more model parameters, etc. However, the computational resources to carry out that scope of a model intercomparison project is not feasible given the multi-million-year simulations we carry out here (the PlioMIP project was focused on snapshot equilibrium simulations with identical boundary conditions) so we can only point out the potential pitfalls of the climate model and parameterizations we employ here. We add in L57-59: “The matrix method interpolation can account for dynamic ice sheet changes like surface lowering, but it does not include changes to paleogeography or ocean circulation”.

In L418-419, we describe our vegetation approach: “The GCM includes a dynamic vegetation module (Kaplan et al., 2003) rather than prescribe Pliocene-specific paleovegetation”; this is consistent with PlioMIP2 (from Haywood et al., (2020): “Model groups could either prescribe vegetation cover from the PRISM4 dataset... or simulate the vegetation using a dynamic global vegetation model.”)

Comparison of ECS – equilibrium climate sensitivity (degrees per CO₂ doubling) – of our model (green; GENESIS (GCM) and RegCM3 (RCM)) compared to the PLIOMIP ensemble including CCSM4-Utr.

Second, the reviewer raises the concern that the modelled EAIS may not require additional physics to collapse [if the climate forcing is not warm enough], and that previous work shows that additional physics are not required to simulate EAIS retreat. While Dolan et al. (2018) do show a suite of ice sheet model experiments characterized by East Antarctic mass loss (their Fig. 5), these snapshots were initialized with a pre-collapsed East Antarctic state – and all models actually regrow from that collapsed state under warm Pliocene conditions – so the simulated East Antarctic mass loss in this specific case is simply predetermined rather than triggered by the climate forcing. Thus, PLISMIP experiments (Dolan et al., 2018; DeBoer et al., 2015) show that warm Pliocene conditions are unable to initiate significant retreat in East Antarctica; ice sheet models with modern initial ice sheet conditions fail to retreat into East Antarctica (Dolan et al., 2018, Fig. 6).

We note that although PLISMIP model members do not trigger EAIS retreat without additional physics (nor do ISMIP6 models; Seroussi et al., 2020, see response to Reviewer 3 for their reproduced Fig. 6), the ABUMIP model intercomparison project (Sun et al., 2020) does show EAIS retreat under idealized forcing. This experiment involves non-realistic extreme forcing (“ABUMIP compares ice-sheet model responses to decrease in buttressing by investigating the ‘end-member’ scenario of total and sustained loss of ice shelves. Although unrealistic, this scenario enables gauging the sensitivity of an ensemble of 15 ice-sheet models to a total loss of buttressing... (Sun et al., 2020)”.

Given that no other models produce EAIS retreat into marine basins (under realistic forcing conditions) without additional physics, we argue that the modeled Pliocene EAIS does require additional physics to collapse, regardless of the climate sensitivity of the climate model. These additional physics, however, do not have to be the MICI mechanisms employed here; for example, Golleger et al. (2019) are able to simulate significant EAIS retreat using a sub-grid ocean melt scheme that applies subglacial melt upstream of grounding lines rather than MICI. Our geologic model/data comparison provides a key validation for the overall sensitivity and spatial response of the ice sheet, with the caveat that mechanisms other than (or in addition to) MICI could be at play.

Using the direct proximal evidence for the Pliocene Antarctic Ice Sheet seems like a good option for evaluating these simulations. However, the threshold between the simulations that are taken as geological constrained and those dismissed is extremely narrow. The two simulations considered as “best-fit” and “highest fidelity” are inconsistent with two of the 10 lines of evidence and a poor fit to two more and you use these to state that the simulations suggest that “the Antarctic Ice Sheet is highly sensitive to past and future warm conditions”. However, one of your simulations (OC3-HFlow-area), with only 5 msle of AIS variability, is inconsistent with two lines of evidence

and a poor fit to three. It seems to me that the available evidence is not of sufficient quality to rule out scenarios based on one extra unclear misfit to the data, especially with the magnitude of the claims that are produced from this. It needs to be made clear in the manuscript that the available data and these simulations really do not rule out a very stable Antarctic Ice Sheet, at least given the assessment made here.

The reviewer brings up concerns that the best-fit simulations are not significantly different than other ensemble members, including very stable ice sheet configurations (e.g., with 5 msle variability from Antarctica). This comment makes it clear that we need to better explain the large differences in terms of quality of the geologic data constraints in different sectors. With that aim, we first clarify the differences between the 'most confident' and least confident' sectors, and then show quantitatively that the best-fit runs score significantly higher because they are more consistent with higher-confidence geologic constraints. This is reflected in the revised manuscript by significant edits to Table 1. (We add to L110-111: "...These geologic criteria, with varied confidence levels based on the robustness of the geologic constraint (Methods), are used to evaluate our ensemble of model simulations (Table 1).")

We now calculate model scores for each ensemble member by assigning values to our previously qualitative categories of model/data agreement (-1=inconsistent; 0=unclear; 1=consistent), and appropriate multiplication factors based on our confidence of the model/data comparison quality for each sector (10=most confident; 5=less confident; 1=least confident). The large differences in these multipliers reflects the large differences in data robustness or appropriateness of model/data comparison. For example, the Amundsen Sea sector with a confidence multiplier of '10' has clearly defined and relatively well-dated evidence of the timing and extent of ice sheet advances across the continental shelf from both seismic and drill-core datasets. The Aurora Basin sector with a confidence multiplier of '1' has directly conflicting data-based interpretations of ice-sheet behavior – we don't know whether the ice sheet retreated deep into these basins in the past, or perhaps didn't retreat at all – and so we definitely don't want to choose our best-fit simulations based on the constraints there. The Pirrit Hills record of ice sheet thickness frequency behavior has a confidence multiplier of '1' because it is probably close to meaningless to compare a mountainside record to model ice thicknesses over a 40-km grid cell. (Although these sectors where model/data comparison is designated 'least confident' are not particularly useful or meaningful when it comes to our goal of constraining model simulations, we feel that it is important to keep these columns in Table 1 because Pliocene geologic records of ice sheet behavior are so scarce, and therefore every little bit of information that can be gleaned is potentially valuable). This information has now been added to Methods, L480-488: "Model simulations are evaluated with respect to each criterion, producing a model-data agreement score ('1' for simulations deemed consistent with the geologic record, '0' for

a poor fit or unclear model-data comparison, or '-1' for a model that violates the geologic record). Each criterion is given a confidence weighting that reflects the strength of the data interpretation or the robustness of the model-data comparison; given the wide range of data quality and ambiguity or certainty around proxies and data interpretation, the weighting factor correspondingly varies exponentially, from '10' (most confident) to '5' (less confident) to '1' (least confident). The model simulation score is calculated from the sum of the model-data agreement scores multiplied by the weighting factor for each criterion (Table 1)."

Now that we have established the differences between 'most confident' and 'least confident' model/data comparison between sectors, the top two (tied) best-fit runs can be defended as clearly better-scoring compared to the 'very stable' (5 msle) simulations, thus supporting our interpretation that "model simulations with the highest fidelity to the geologic record show large-scale [Antarctic ice sheet] fluctuations."

(Note: The simulation mentioned above, OC3-HFlow-area, with two 'inconsistent' scores and poor fit to three, actually has 14 msle of variability, and deglaciates somewhat into East Antarctic basins, so would not be characteristic of a 'very stable ice sheet')

Revised Table 1:

There are a couple of significant changes that need to be implemented for this manuscript to be suitable for publication, namely:

- The close match between the geological record and simulations with very little ice sheet fluctuation needs to be recognised much more in the manuscript. Particularly, this needs to be included in the summary statement and the abstract. I would recommend finishing the final sentence of the summary with the clause “without ruling out very stable ice sheet parameter sets”. In the abstract, I would rewrite the sentence starting on line 11 to be “Geologically consistent simulations reveal a dynamic ice sheet that could have contributed up to 25m of sea level change across glacial-interglacial cycles, although very stable ice sheet configurations cannot be ruled out.”

We have now quantified our previously qualitative rankings. With this additional clarity, we note that the three ensemble simulations that are ‘very stable’ (e.g., produce 5 msle of sea level fluctuations) systematically violate or produce a poor fit to three out of the five ‘more-confident’ sectors (i.e., the sectors with robust geologic data constraints), and produce significantly lower run scores than our best-fit simulations. We therefore add to the abstract (L14): “Geologically consistent simulations reveal a dynamic ice sheet that could have contributed up to 25m of sea level change across glacial-interglacial cycles, while relatively stable simulations produce worse fits to data constraints.”

- The climate model simulations that make up the matrix driving the ice sheet model needs to be included in the main text, somewhere around line 48. This description should include the CO2 levels, orbital configurations (including times represented) and Antarctic Ice Sheets incorporated (maybe with an image of the latter). It should also mention the things that aren’t included, especially Pliocene palaeogeographic changes, climate feedbacks from EAIS loss and changes in ocean circulation.

This information has now been added to the main text (L51-54).

(Note that climate feedbacks from EAIS actually are represented in the matrix method, since the ‘collapsed-WAIS’ climate snapshot also includes ice loss from EAIS marine basins. This is also now stated on L410).

There are also a number of changes that would improve the manuscript:

- The opening sentence of the abstract makes it sound like the Pliocene is rarely as warm as modern, whereas temperature records suggest that it was warmer for very long periods of time, at least 100s of thousands of years, if not millions of years. Please rephrase to reflect this.

Rephrased: “Ice sheet dynamics during the Pliocene, when global temperatures periodically exceeded and remained elevated above modern conditions...”

- Line 5 of the abstract states that you are using coupled ice sheet and climate models. This is only true in the most generalised of senses and not in the normal scientific meaning of this phrase. The following would be a much clearer sentence to include

here. “Here we employ an ice sheet model with forcing provided by a matrix of climate model simulations to simulate transient Antarctic ice sheet cyclicity from 4.5-2.6 Ma. This provides ...” *Rephrased accordingly.*

- Line 8: From this manuscript I cannot see how the data is used to “constrain the model” or “ensure geologic compatibility”. Better verbs would be to “evaluate” and “test” respectively. *Corrected.*

- Line 10: I do not see any point in the manuscript where the Antarctic geological record is extrapolated. Either remove this sentence or make clear what this extrapolation is. *Replaced ‘extrapolate’ with ‘translate’; e.g., “This model-data comparison approach is used to translate the intermittent geologic record into a continuous reconstruction of Antarctic contribution to sea level.”*

- Line 26: Do you mean the Pliocene sea level signal? It currently reads as the future sea level, since this is the last sea level record you mention, which would make no sense. *Yes; rephrased accordingly.*

- Line 31: using a constant climate could melt the ice sheet rather than build it up, just depending where in a climate cycle is simulated. *Added this consideration.*

- Line 32: Many modelling studies (e.g. PLISMIP) incorporate the uncertainty due to hysteresis (and it is relatively straightforward to do so). Maybe add “although modelling can take this into account” at the end of the sentence. *Changed to: “the equilibrium snapshot method can introduce additional uncertainty due to hysteresis in initial conditions (Dolan et al., 2018).”*

- A better description of how subsurface temperatures are calculated should be included in line 49. *Changed to: “Ocean temperatures are scaled from a modern climatology using the matrix method weighting scheme to either apply a uniform ocean temperature anomaly for warmer-than-present times, or interpolate between a modern and glacial ocean for colder-than-present times (Methods).”*

- Line 139: I think it is important to point out here that although your model cannot retreat into the Wilkes Subglacial Basin without MICI that other models are able to reproduce this, albeit with different climate forcings (e.g. PLISMIP)

To our knowledge other models cannot simulate retreat into the Wilkes Subglacial Basin or other marine East Antarctic basins without MICI or a sub-grid ocean melt scheme under realistic forcing conditions (see response above).

- Section 2.3: Please could you explain why warming episodes in your model setup do not increase the height of the Antarctic Ice Sheet, through increased precipitation. Is this something the model is unable to do or does not reproduce? This relationship is attested by present day observations, future ice sheet simulations and Pliocene climate and ice sheet modelling, so why does this not hold for your modelling setup?

Yes, warming episodes in our model do increase the interior height of the ice sheet (this behavior is later discussed at length in Sect. 3.4). Here in Sect. 2.3, we now clarify: “...in these simulations, inland thickening occurs during interglacials due to precipitation,”

while coastal thickening occurs during glacials due to marine ice growth.”

• Line 196: It is fair to say that there are significant sources of uncertainty in oxygen isotope based sea level estimates, but I am a little uncomfortable dismissing them as simply unreliable. Maybe something like “though $\delta^{18}\text{O}$ estimates have significant uncertainties, particularly when the multiple factors that can affect the signal are considered¹⁶.” *Corrected to “...though $\delta^{18}\text{O}$ estimates of sea level have significant uncertainties.”*

• Line 306: I am confused about your comments on the mPWP, given that the results presented here show the highest CO₂ concentration (and therefore presumably temperatures) and the longest period of retreated Antarctic Ice Sheets during the mid-Pliocene warm period. It is very plausible that the early Pliocene was warmer than the mid-Pliocene, which is suggested by long climate records, and especially so in the Southern Hemisphere when the Isthmus of Panama was open (although you wouldn’t capture this here). *Removed this sentence; also removed Conclusions L349-350 accordingly.*

• Line 322: I don’t know of anyone who suggested that the mPWP was the warmest period of the Pliocene, globally or in Antarctica. So please give a reference here, if you are trying to refute something in particular. In fact many people have suggested the early Pliocene was significantly warmer than the mid-Pliocene (e.g. ANDRILL or Ohneiser et al., 2020).

In the revised manuscript, we adopt a more nuanced tone in this manuscript regarding the MPWP (see L342-343, and edits to Conclusions as well, L360-361)

• Line 378: I don’t think it makes any difference, but it is worth noting that CCSM4-Utr is one of the warmer models in PlioMIP2 and by some estimates has the strongest response to non-CO₂ Pliocene boundary conditions (Haywood et al., 2020). *Added (L428-429)*

• Line 385: It should be noted that modern melting of Antarctica is driven by upwelling of deeper waters penetrating into glacial troughs on the Antarctic shelf (Walker et al., 2007; “Oceanic heat transport onto the Amundsen Sea shelf through a submarine glacial trough”). So I don’t think this formulation would do a good job of simulating observed melting of Antarctica, as it is not closely related to the mixed layer temperatures. Is this something you have considered?

Yes, this is the reason that we take the extra step of relating mixed layer temperatures to 400-m layer temperatures (responsible for melting at grounding lines) using the coupled CMIP model output in Fig S7 (rather than just using ocean temperature anomalies from our mixed-layer ocean model).

• Line 393: You have data from the CCSM4-Utr Pliocene model, so please justify why the same scaling was not applied when temperatures were above modern. *Added explanation.*

• Line 559: Please note here that there are marine sediments above the modern ice

surface, which suggests significant topographic change that is not included in these models.

The model includes bedrock isostatic response to ice sheet load changes; while we didn't explicitly require that these locations be underwater (e.g., marine) in the Pliocene due to the coarse grid resolution in our model, this region does experience significant topographic change across glacial/interglacial cycles.

- Line 599: should say “model-data” rather than “mode-data”. *Corrected.*
- Table 1: “Sea level amplitude” is a good metric, but most attempts to produce a Pliocene sea level estimate aim at maximum sea level. It would be great to have another column with the maximum sea level in metres above present day sea level, so the simulations can be compared to data. *Added as a column to the revised Table 1.*
- Figure M3 is an absolutely key figure, in that it shows the forcings that go into the model, so needs to be in the main paper. It could perhaps be added as panels in Figure 2. It would also be great to have the three orbital forcing configurations that are actually simulated added to panel c with a symbol. *Added to Figure 2 as suggested; additionally, the end-member orbital forcing configurations (and CO₂ concentrations) that we used for our climate model matrix are denoted in Fig. S5 rather than Fig. 2 for simplicity.*

Reviewer #2 (Remarks to the Author):

Review Halberstadt et al.

GEOLOGICALLY CONSTRAINED ANTARCTIC ICE SHEET SIMULATIONS: 2 MILLION YEARS OF GLACIAL RETREAT AND EXPANSION THROUGH THE WARM PLIOCENE

Halberstadt et al. present an ensemble of paleo ice sheet simulations spanning the time from 4.5 – 2.6 Ma BP. Employing a matrix/climate index offline forcing approach and a coarse resolution model setup (40 km) they assess the evolution of the AIS during glacial-interglacial cycles and find extended periods of retreated ice sheet configuration driven by MIS1 and MIC1 but also individual periods in which the AIS attains a full glacial state. They calibrate and discuss their model results in light of the available proxy data finding good agreement with i.e. marine sediment cores suggesting sustained ice free conditions in different ocean sectors. They conclude that an intermediate parameterization of MIC1 in PSU-ISM yields the best results with respect to the proxy records, showing that low MIC1 parameters do not produce sufficient ice retreat compared to proxies and high MIC1 does not allow for an expanded ice sheet margin. Both cases are not in line with current estimates from paleo-proxy reconstructions. This

is an important step in calibrating a highly uncertain process such as MICI in an ice sheet model. The manuscript is very well written and easy to follow despite the complexity of the approach and the various proxy records taken into account. The method section is substantial and very well written shedding light on the more involved aspects of the model setup employed. The main flaw of the study is the model resolution (40 km) which is very coarse for current standards. While I realize that simulations covering such a long time-span can be prohibitive from a computational standpoint it should be possible to at least produce a higher resolution run of the best fit ensemble-members highlighted in table 1. In his highly cited paper from 2009 Pollard and DeConto (both co-authors here) carried out 5 Ma of model simulations on 40 km including nested 10 km res simulations as well. I am aware that model physics were probably not as complex as the current version of PSU-ISM but say 15-20 km res simulations for a subset (see above) should be manageable and would improve both the quality of the paper as well as the robustness of the results. The second aspect I was missing is a PD or PI calibration of the model setup. As the authors correctly state, proxy recs for the Pliocene are sparse and sometimes of qualitative nature while we have a very good coverage of the current AIS state. I am aware that it is almost customary to reference previous instalments of the model stating that calibration has been carried out there and the model performs acceptably under well constrained reference conditions. It would be nice to see some more information on that in the supplements i.e. equilibrium thickness change and grounding zone positions for e.g. PI-cntr conditions.

Thank you for your constructive feedback to improve our work. In response to these suggestions, we have added two supplemental figures. First, we show our PI validation of the model setup (Fig. S4, reproduced below), and referenced on L392. For brevity, only a representative subset of the ensemble is plotted in Fig. S4, but all parameter combinations maintain a reasonable modern grounding line after 2000 years under constant preindustrial conditions (e.g., modern orbital configuration, CO₂, and ice-sheet topography).

Fig. S4. Model simulations under constant preindustrial matrix-interpolated climate. Each parameter combination in our ensemble is tested under preindustrial conditions. Simulations are run for 2000 years with constant modern orbital parameters and CO₂ concentrations set at 280 ppm, and the resulting ice thickness and grounding line is compared to Bedmap2 (Fretwell et al., 2013). Only a subset of model parameter combinations is shown here; (a) Initial restart file; (b) OC3-HFlo-area; (c) OC3-HFoff-vol; (d) OC3-HFmedlo-area; (e) OC4-HFmedlo-vol; (f) OC4-HFmedhi-vol; (g) OC3-HFmax-area; (h) OC3-HFmedhi-vol.

We also demonstrate that our results are insensitive to model resolution; we have performed higher-resolution (15km) nested model simulations for a best-fit run, and show that the grounding-line patterns across the continental shelf in this 15km-simulation are quite similar to the continent-wide model. Since the patterns of ice advance and retreat across the continental shelf provide the model-based criteria for the majority of our data constraints, we therefore conclude that our results are insensitive to the necessarily coarse resolution of our continental ensemble (Fig S2; and discussed in L397-398).

Fig. S2. Insensitivity of results to model resolution. Higher-resolution (15km) nested simulations, driven by boundary conditions from a best-fit continental run (40km resolution), produce very similar grounding-line fluctuation patterns (a,b) and extents (c). (a,b) Grounding-line behavior at each timestep is plotted with respect to distance from the continental shelf edge (black star) along the trackline shown in the left-hand-side plot (dashed black line) in the (a) Ross Sea and (b) Amundsen Sea. (c) Nested simulations produce the same grounding line positions through time (right) as the continental simulation (left). Grounding line color indicates the ice sheet grounded area with respect to the modern extent.

This being said I congratulate the authors on an interesting study and I would suggest publication in light of my points above and some minor comments outlined below.

Best Regards,

Johannes Sutter

L70 .. “Best-fit model simulations are used to extrapolate pinpoint geologic records, disparate in space and time, into a continuous and geologically constrained reconstruction of AIS contribution to Pliocene sea level.”

This is a nice idea, what about uncertainty assessment due to ISM and boundary conditions?

Best-fit models robustly extrapolate the geologic record into a spatial reconstruction of AIS evolution (which translates to a sea level contribution), broadly independent of these uncertainties.

Uncertainties around ISM physics and parameterizations, boundary conditions, etc. come into play more strongly in the next step, when we use geologic constraints to try to identify discrete parameter ranges / specific tipping points for the ice sheet, because those specific values are dependent on the ISM and boundary conditions used. (See response to Reviewer 3 for more detail).

L93 “trough mouth fan development”

maybe provide a reference as this manuscript is intended to be understandable by a wider (non-expert) audience.

Replaced with: “...the accumulation of glacially-triggered debris flows on the continental shelf slope during the Pliocene suggests that the ice sheet periodically advanced to the shelf break”

L188 “GMSL ranges from +18m during an Antarctic interglacial/Greenland glacial period (+18m from the Antarctic Ice Sheet plus 0m from an approximately modern-size Greenland) to -2m during an Antarctic glacial/Greenland interglacial (-7m from Antarctica plus +5m from Greenland), resulting in GMSL amplitude of 20m.”

I don't quite understand the reasoning here. During a period of NH glacial advance (I'm not sure whether the term Greenland glacial period is fortunate here as globally your either in an interglacial or glacial but not both at the same time) antiphased to SH retreat you can't exclude a Laurentide or fennoscandian component growing so the assumption +0m is a bit unfounded. Likewise, why capping the max Greenland contr. to 5 m, PD GIS has about 7m SLE.

(1) Rephrased glacial/interglacial language here to ‘ice sheet growth/retreat’

(2) Here we assume that there were no large-scale Northern Hemisphere ice sheets growing during the Pliocene that could have masked AIS sea level contributions, just Greenland; we now make this assumption explicit (L207-209). (However, our ongoing and future work investigates this assumption!)

(3) We cap the ‘Greenland retreat’ sea level contribution at +5m because the “Greenland ice sheet contributed up to ~5-7 m sea level equivalent during the Pliocene (ref 13: de Boer et al., 2017)” (L201 has now been edited for clarity). However, we feel it is equally defensible to use the 7m Present Day ice storage of GrIS, as suggested

above; we have redone the calculations accordingly (the resulting GMSL amplitudes are 18m (antiphased) and 32m (phased)).

L211 “Specifically, zero or low MICI parameter values cannot generate enough grounding-line retreat across Wilkes Subglacial Basin (to produce large pulses of far-traveled iceberg rafted debris) or Prydz Bay (to deposit inland open marine sediments),”

Couldn't an expanded Wilkes Subglacial Basin ice cover (grounding line adv. + ice thickening) which quickly retreats to PD conditions also produce a large far traveled iceberg armada signal). At least this would be an alternative explanation which might be as conceivable as a large inland retreat of the grounding zone.

The goal of this work is to incorporate the original data-based hypotheses as faithfully as possible; the primary literature interpreted the IBRD signal as reflective of “large-scale destabilization of the East Antarctic Ice Sheet” (Cook et al., 2014). I believe this interpretation is based on both the observation of IBRD pulses occurring during very warm interglacials, as well as the provenance of IBRD (sourced from rocks inland of the modern coastal region).

L219 If MICI is a process which has driven ice sheet instability in the past, a valid parameterization for these past climate states (both on the warmer and colder spectrum than PD) should also be valid for projections such as done e.g. in DeConto et al. 2016. Otherwise, one would have to change parameterisations of MICI depending on the climate state which would not be very helpful I guess. Wouldn't it be possible to at least inform the parameter range which DeConto et al. used and therefore reduce the associated uncertainty in hindsight?

Fig S3 places our best-fit MICI parameter values within the context of the DeConto et al. (2021) future projections (RCP 8.5 sea level by 2500). We chose not to explicitly constrain the uncertainty in future projections because our parameter variations don't thoroughly sample the parameter space (specifically between VCLIFF=3 and 13), but Fig S1 can be interpreted accordingly. For example, none of our simulations with 'max' MICI parameters (VCLIFF=13 and CALVLIQ=180) were consistent with the Pliocene geologic data; therefore, our work indicates that ~25m of esl from Antarctica by 2500 is an overestimate. This is consistent with the best-fit Pliocene simulations producing a maximum of 22m esl above present.

L226 “Ice sheet fluctuations are driven by variation in climate forcing” as well as internal dynamics/feedbacks?

Replaced with “Ice sheet fluctuations are driven by variation in climate forcing acting in tandem with internal feedbacks.”

L285 “In other words, our simulations indicate that the ice sheet thickening at coastal locations reported in Yamane et al.50 occurred during Pliocene glacials, while thickening at interior locations occurred during Pliocene interglacials. During these interglacial warm periods, however, ice sheet contribution to GMSL during warm periods was overwhelmingly dominated by mass loss at marine margins despite increased surface accumulation.”

I guess this is somewhat to be expected as it basically corresponds to late Quaternary glacial interglacial thickening/thinning patterns in the interior EAIS where both ice core proxies as well as modelling suggest thickening during interglacials and thinning during glacials due to SMB changes.

Replaced this sentence to better reflect the novel contribution of our work: “... our simulations provide temporal context for the various locations where Yamane et al. reconstruct higher-than-present ice elevations: thickening at their coastal sites occurred during Pliocene glacials, while thickening at their interior sites occurred during Pliocene interglacials, similar to ice thickness changes in the late Quaternary. Despite increased surface accumulation during warm periods, however, interglacial ice sheet contribution to GMSL was overwhelmingly dominated by mass loss at marine margins.”

L322 “Antarctica but earlier than the MPWP, suggesting that the globally recognized MPWP may not have been the warmest Pliocene period in Antarctica”

Did you test your model setup on the MPWP, would be interesting to see your model-response in comparison.

We did not perform MPWP equilibrium simulations following the PlioMIP protocols; however, our model simulations do span the MPWP (Fig. 2). Due to the elevated CO₂ forcing during this time, we simulate a time of extended warmth. Here in Sect 3.5 we contextualize this period of MPWP warmth with the rest of the simulated record of continental shelf ice sheet expansion, and notice an earlier period of extended ice sheet retreat from the continental shelf.

L324/325 You mentioned before that it is not straightforward to assess the DeConto et al 2016 parameterization with your approach here, but it would be still nice to contextualize this at least approximately. E.g. the MICI parameterization is on the lower/higher/mid range compared to what’s been used the 2016 paper.

L362: “Our model-data comparison indicates that only intermediate values of modeled MICI parameters are consistent with the geologic record...” Here and elsewhere, ‘intermediate’ describes the MICI parameterization with respect to previous work (e.g., DeConto & Pollard 2016, DeConto et al 2021).

Methods:

L336 I suggest to specify this a bit more, acceptably is a very subjective term in this context. To see caveats it would be good to know whether e.g. grounding line reversibility etc. is comparable to the mean model response or belongs to the better/weaker performing models.

Edited to: "...with a grounding-line ice-flux formulation that demonstrates grounding-line reversibility and reproduces theoretical and full-Stokes modeled grounding-line behavior in idealized model intercomparison studies."

L340 what parameterization is used for sub-ice basal melt rates and what is the reasoning for the scaling parameters? I am aware that this is probably mentioned somewhere in ref 2 but a short sentence here would be helpful.

Edited to: "...multiplying the sub-ice melt rates (parameterized as a quadratic dependence on temperature following Martin et al., 2011)."

L341 maybe I'm reading this wrong, but does that mean, ice cliffs retreat 100 km, 200 km, 300 km or 1200 km per year? I'm sure this is not the case but for readers which are not familiar with the MICI parameterization in PSU-ISM it would be nice to quickly specify what this means in real wastage rates and under which conditions these are switched on.

VCLIFF wastage rates are km/yr; our parameter variations span 0 – 12km/yr (the typo here has been corrected). We also add: "for context, terminus velocities at Jakobshavn Isbrae have been measured up to ~12 km/yr (Joughin et al., 2012)"

L349 It would be great to see some delta plots (e.g. deltaH) for PD/PI conditions and e.g. ice extent for LGM in the supplements. This gives the reader a clearer idea about potential biases in the modelling approach.

Added Fig. S4.

L352/353 it would be good to have some sense of resolution-sensitivity. I suggest to run a shorter timeframe (couple of glacial-interglacial cycles e.g 200 kyr) on e.g. 15-20 km to see how the model response will vary. This should be possible computationally for a subset of parameterizations (e.g. min/max and middle of the road). Alternatively a nested approach as in Pollard & DeConto 2009 could be reasonable.

Added Fig. S2.

L354 This means there's no model spinup? Probably not a larger issue due to multi-

million year period you are investigating but would be good to mention how the initialization shock looks like.

Initialization shock lasts <5kyr and, for most (but not all) parameter combinations, mostly reflecting ice sheet growth due to increased precipitation during the Pliocene.

L423-426 the geologic constrained compiled by you for this study could be a very valuable resource for the community in general if available from a public repository. If all the data is published somewhere and can be uploaded this would be a great bonus outcome of this study.

We hope that this resource will have continued utility for the wider community. For each region, we list the datasets and references compiled for each region, along with the explicit model evaluation criteria developed here based on this data compilation.

L427-430 I think a similar sentence should appear in the main part of the manuscript so the reader can interpret the results with all biases taken into account.

Added L111-115: “The computational effort of performing an ensemble of multimillion-year simulations requires a relatively coarse (40 km) model spatial resolution, so our interpretation of the geologic record and model-data comparison efforts are correspondingly large-scale; however, higher-resolution nested simulations demonstrate similar patterns of grounding-line fluctuation (Fig. S2).”

L508 How well can ice retreat be quantified from these sediment records as opposed to just saying there was retreat but we don't really know the baseline state of the ice sheet neither how much grl retreated. What I am getting at is whether these records are robust enough to make a statement about MICI parameterization. I assume there is grl retreat into the Wilkes Subglacial Basin and Aurora basin also without MICI, what is the reasoning behind that this is too small of a change (maybe I have missed this in the main manuscript).

While we have a relatively clear boundary on the maximum extent of collapse, from the ϵ_{Nd} data precluding extensive grounding line retreat into the Adelie Craton region, we agree that it is difficult to quantify the grounding line position that could have produced the sedimentologic signal of “large-scale collapse” from the Wilkes Subglacial Basin.

We therefore split the Wilkes data into two categories and assess them separately: the more clearly interpretable ϵ_{Nd} provenance constraint on max grounding line retreat (‘WSB’ under ‘Extent of ice retreat’ category in Table 1); and the sedimentologic/IBRD evidence for “large-scale collapse” (‘WSB’ under ‘Ice advance and retreat pattern’ category in Table 1).

Regarding the sedimentologic/IBRD evidence (i.e., sedimentation patterns indicating a smaller-than-present ice sheet, and evidence of far-traveled iceberg rafted debris sourced from this region): because of the lack of a precise spatial extent of retreat, we take a conservative approach and interpret any model simulation with a grounding line receded from the modern position by about 200km (basically, just enough to form a slight grounding-line embayment), as consistent with the geologic record. Because this is such a small change, most of the model simulations in the ensemble are ranked 'consistent' (green in Table 1) in the WSB 'Ice advance and retreat' column. This particular constraint just penalizes a few model members: those with high ocean sensitivity (OC4) where the simulations never advance across the continental shelf as indicated by the sedimentary record, or those with no or low MICI that do not retreat beyond the modern configuration far enough or frequently enough to match the IBRD record.

Is this record "robust enough to make a statement about MICI parameterization"? As described above and shown in Table 1, we interpret this record conservatively enough that it doesn't have a lot of resolving power on its own, but in combination with other datasets in other regions (e.g., the Prydz Bay data constraining min extent of retreat, and the ϵ_{Nd} dataset constraining max extent), we can narrow down the range of MICI parameters that fit the geologic data as a whole. This demonstrates the value of considering a suite of geologic records from around the continent, rather than focusing on just one region.

L522 It is impossible to adequately resolve the transatlantic mountain range at the resolution considered here. There will always be too much ice accumulating as small outlet glaciers are insufficiently resolved and ice drainage is too slow. This is a typical problem with coarser resolution continental scale simulations so I'd be very careful interpreting results from this region.

This is a longstanding issue in continental models of Antarctica. To compensate for this, we apply an ad-hoc adjustment that scales the basal sliding coefficient based on the subgrid roughness (the standard deviation of bed topography within a model grid cell, based on Bedmachine). This tuning is described in Pollard & DeConto (2012) and allows closer-to-observed flow across the Transantarctic Mountains in modern simulations.

Table 1 it would be interesting to disentangle in the matrix which element has low confidence/poor fit due to known model biases. As you mention in the manuscript, comparing ice thickness changes at Nunataks at this resolution is not very informative, and I'm wondering why one would include it at all in this assessment.

We include this equivocal model-data comparison purely for the sake of completeness – since the available data are so limited, including everything (even if the model-data comparison is not very informative) highlights both the need for more data collection as well as the comprehensive nature of our geologic compilation. The nunatak column in Table 1 has now been shrunk significantly, which should further highlight the equivocal nature of this model-data comparison point (as should the explicitly quantified confidence weights we have now added).

Figure 1, 2, 5: the 2d AIS plots are cut of at a latitude which masks the Peninsula and parts of the EAIS. I suggest to make the cut at a lower latitude.

Our model domain for the continental runs is quite restricted due to computational expense of these multimillion-year simulations – the established domain is as small as possible while still providing robust representation of the ice sheet (see below) – so plotting these domains at a relatively high latitude encompasses most of the domain and highlights the areas where we performed model-data comparison (we did not use data from the Antarctic Peninsula here).

The model domain plotted with a lower-latitude cut.

Figure 1 and 2: from the time series it seems that some glacial-interglacial-glacial transitions don't show hysteresis (i.e. fast retreat, slow regrowth). It would be nice to hear the authors view on why this is (forcing, model parameterization, combination of the two)?

We had originally elaborated on this interesting observation, but decided to remove this discussion from an earlier manuscript draft both due to length constraints and also because we plan to pursue this further in future work!

Figure 3: this is a nice way of visualizing the results. Maybe add cm/yr to the colorbar label.

Added.

Figure 4: while this is a nice way of illustrating the results I think one could move it to the supplements. It is clear from a physical point of view that deglaciation rates in cases of ice sheet collapse are much faster than growth rates (the latter being mainly fueled by a cessation of high surface and basal melt rates and not by nonlinear effects such as MISI and MICI). Seeing this quantified in a figure is nice but does not really show anything new per se.

The new information that we intend to convey with this figure is allowing the reader to visually locate the specific tipping point thresholds specified in the main text. We agree that this is our least important figure, and it will be the first to move to the supplement as length limits require.

Figure 5 I'm a little confused by the figure caption (d) your model results show interior EAIS thickening in interglacials and in the manuscript you attribute this to higher precip rates which makes sense and is what we see in late quaternary interglacials. However, in the caption of (d) you state, that your model produces higher interior precip rates for cold glacial orbits (keeping CO₂ at the same rate). Is this representative of your glacial states in the matrix method or just a special case? If it is representative than I don't understand the thickening in interglacials even though integrated SMB would be smaller. Please clarify and modify the respective paragraphs if needed.

Thanks for catching this error – this was a 'typo' and has been corrected.

Figure 6 I realize that this is supposed to be a qualitative figure, but it is difficult to disentangle what it illustrates. To begin with, the continental shelf edge is relatively close or far away for PD reference conditions. Is the vertical axis calibrated with respect to this? Is distance measured along latitude or nearest point? Does the lower limit of the y-axis represent the continental shelf edge and the upper limit is positioned according to scale? If so the lower end of the y-axis would be zero right? It would help to add y-axis ticks as well. From an aesthetic point of view, inlet (c) could be improved.

The revised figure indicates the modern (PD) conditions along transect (dashed black line), and each transect is now labeled on the spatial map and corresponding plot.

(Revised Figure 6): Modeled interval of prolonged mid-Pliocene ice sheet recession in all major catchments. Model grounding line position (OC2-HFmedhi-vol) in each major Antarctic catchment is plotted along a continental shelf transect indicated in the spatial plot. The modern (modeled) grounding line in each catchment region is denoted by a black dashed horizontal line. Yellow shading indicates the “Pliocene Amundsen Sea Warm Period”, a time when the ice sheet was primarily receded in the Amundsen Sea (4.2-3.2 Ma, although the modeled warm period begins a bit later at ~4.1 Ma). A similar period of prolonged mid-Pliocene ice sheet recession is evident in other catchments around the Antarctic Ice Sheet during this time. This time period coincides with the time of greatest warmth in the Ross Sea AND-1B record.

Reviewer #3 (Remarks to the Author):

Halberstadt et al. present model experiments to simulate the transient behaviour of the Antarctic ice sheet through the Pliocene (4.5-2.6 Ma). They also present a compilation of geological evidence to document advances and retreats during this time period. These geological evidence are used to constrain their model results. These simulations are clearly interesting as they intrinsically represent the transient nature of this time period while the majority of previous modelling works have focused on selected boundary conditions supposedly representative of typical “Pliocene” climate (e.g. warm/cold orbit, high/low co₂, etc.). I generally enjoy reading this paper as I think that the chosen methodology is sound and the results provide a better understanding of the Antarctic ice sheet dynamics through the Pliocene. However, I have a few comments that might deserved some discussions.

We thank Reviewer 3 for their helpful feedback.

Major comments

1. Uncertainties and model constraints. I appreciate the authors' effort to compile available data to constrain their model results. That being said, I found the "constraints" offered by this data relatively weak as they don't offer precise indication on space nor time. Since these constraints are so weak I feel that many model experiments could in principle agree with them.

Some of the geologic data constraints are indeed weak; we have now further clarified the differences between these weaker data constraints (labeled as 'least confident') and stronger datasets ('most confident') in our scoring scheme. We use these confidence categories to ensure that our model-data comparison results are primarily influenced by the most robust datasets.

By nature, these geologic datasets do not offer precise indications in space and time (e.g., the ice sheet grounding line reached 73°S 176°E at 4.2011 Ma), but they do contain valuable information that can be assessed in space and time (e.g., there were at least 13 glacial advances across the ANDRILL-1B drill site in the Ross Sea after 3.4 Ma). This is the first work, to our knowledge, that has attempted to explicitly integrate all currently available proximal geologic data to inform ice-sheet modeling. Some datasets have less constraining power (e.g., for Weddell Sea: at least one grounding line advance to the shelf edge), but even so, not all model members satisfy this criterion. Other datasets have more constraining power (e.g., no grounding-line retreat into Adelle craton region), and many model members violate this constraint.

The authors here have explored only a very restricted model parameter ensemble (sensitivity to oceanic temperature change, MICI and a methodological parameter). I feel that the ice sheet dynamics shown here is affected by some modelling choices: the ice sheet model used, the climate forcing used to generate the climate matrix, the parameter in the regional climate model used,... On top of that the greenhouse gas concentration reconstructions is far from being perfect and is associated with a considerable uncertainty. I do not expect the authors to quantify explicitly all these uncertainties as the simulations presented here are probably numerically expensive. However, after reading the paper, I have the feeling that one could have simulate different sea level amplitudes and different waxing/waning frequency using a slightly different setup. Perhaps at least a discussion on model biases and how these biases could affect the results could be added in the manuscript? And the same for the CO₂ curve?

Due to computational constraints, we explore only a restricted number of model parameters, as noted here. Our goal for this restricted ensemble was to simulate different sea level amplitudes and different waxing/waning frequencies within this parameter space, in order to see what patterns of esl / ice sheet fluctuation frequency best match the geologic data.

The considerable uncertainty associated with our modeling approach impacts various aspects of our results differentially. In this work, our primary goal is to simulate geologically-consistent ice sheet evolution across Pliocene glacial/interglacial cycles. We take our ‘best-fit’ simulation(s) and make interpretations about the AIS contribution to sea level, the spatial extent of inland retreat, etc. These results are less sensitive to model uncertainties because they are built only on our observations of modeled ice sheet behavior in the simulation that best reflects the geologic record – so, regardless of the model climate sensitivity or potential CO₂ errors or incomplete model physics, if a simulation matches geologic records through space and time around the Antarctic continent, then we can use this simulation to reconstruct time-evolving AIS behavior and make interpretations about sea level contributions.

Model uncertainties pose a much larger challenge when we use these ‘best-fit’ simulations to back out model parameters or discrete climate variables, for example, when we identify temperature and CO₂ thresholds for ice sheet collapse. When used for this purpose, we agree that our results need more discussion of uncertainty – a climate model with a different climate sensitivity might result in a higher or lower CO₂ threshold for ice sheet collapse, for example, and these thresholds may also depend on inaccurate or poorly parameterized model physics. We have added caveats in key locations where these uncertainties need to be considered:

L251-252 (regarding the MICI parameter constraints indicated by our model-data comparison): “...our model-data comparison using spatial geologic information provides an upper and lower limit on geologically consistent MICI parameter values (Fig. S3), although the exact values identified here are specific to this ice sheet and climate model setup with associated uncertainties.”

L264-266 (regarding the identification of temp and CO₂ thresholds for ice sheet collapse): “The specific thresholds of these forcings vary for each simulated collapse event and are generally characterized as >400 ppm CO₂ and >-20 W/m² in our simulations (Fig. 3a), although the precise values of these thresholds are model-specific and depend on climate model sensitivity and model parameterizations.”

We elaborate further in Methods, L468-469: “The timing of ice sheet grounding line fluctuations is sensitive to this highly uncertain paleo-CO₂ formulation, though the amplitude is robust (model simulations across a time interval where orbital-scale reconstructions are available (3.3-2.6 Ma) produce similar amplitudes of glacial cyclicality

as models forced by the $\Delta\delta^{13}C_{P-NA/2}$ CO₂ proxy). Although our $\delta^{13}C$ -derived CO₂ proxy varies widely, model ice sheet behavior does not directly mirror the CO₂ time series forcing (Fig. 2a,b)....”

2. Sensitivity to atmospheric and oceanic climate change. From the results presented it seems that without the MICI (HF off) the model is unable to simulate grounding line retreat in East Antarctica. From Fig. 1c it also seems that no ice volume lower than PI is simulated in HF off.

This comment has prompted the addition of a supplemental figure showing the time-evolving ice sheet area (below; y-axis is ‘percent of modern grounded area’), not ice sheet volume (esl, as in Fig. 1). Here we additionally divide the behavior of the WAIS (b) and EAIS (c), with the total AIS in (a). Now that we can view our results in terms of grounding-line retreat rather than just sea level contributions, we demonstrate that the model does simulate some East Antarctic grounding line retreat even with HF off (blue lines in c below).

Fig. S1. Model ensemble runs, colored by hydrofracture parameter combinations, are plotted with respect to grounded ice area change compared to modern for (a) the entire ice sheet, (b) WAIS only, and (c) EAIS only.

For the forcings representative of the Pliocene we can expect some ice sheet retreat in East Antarctica without invoking the MICI. For example in recent intercomparison exercise for future warming scenario in ISMIP6 (Seroussi et al., 2020) or for the buttressing intercomparison exercise (Sun et al., 2020), some models produce East Antarctica grounding line retreat. The model used here might present a relatively weak sensitivity to the oceanic forcing.

In the new supplemental figure above, we now show that the models with no MICI (HFoff) can simulate grounding line retreat beyond modern in EAIS basins (Fig. S1c). A spatial example is provided below, comparing a no-MICI model result for a warm Pliocene time period (black) to the modern grounding line (red)

Our no-MICI model actually produces more EAIS grounding-line retreat than ISMIP6 projections for 2100 (Seroussi et al., 2020, Fig. 6 reproduced below):

REDACTED

Indeed, this no-MICI warm-Pliocene grounding line result is more on par with ABUMIP (Sun et al., 2020, Fig. 4 reproduced below), despite their non-physical experiment setup (designed to highlight the differential response of models under extreme idealized forcing). This reflects that Pliocene conditions far exceed the 2015-2100 projected range of ISMIP6.

REDACTED

In summary, our model is responding to ocean forcing in a similar manner as previously published model intercomparisons.

Also, I was wondering about the atmospheric sensitivity. There is no information on surface mass balance computation in the manuscript (PDD? lapse rate?...), only precipitation changes are discussed. With the forcings used, do you simulate surface melt? Does it explain part of the grounding line retreat simulated?

Yes, much of the MICI-driven retreat is triggered by surface melt. We have now added this information to Methods. L393-394: “Surface mass balance is calculated using a positive-degree-day scheme, with a lapse rate correction for topographic differences.” And we now explicitly specify in L385: “...surface meltwater availability...”

3. Early Pliocene ice sheet volume. I was surprised to see a systematically larger ice sheet than today for the earlier part of the record (Fig. 1 and Fig. 2). Is this linked with the CO₂ reconstruction based on delta¹³C? For this part of the simulation, we can expect a role of Northern Hemisphere ice sheets as the climate seems favourable to their growth. I do not think that Northern Hemisphere ice sheets for this time period are supported by geological evidence... More generally in the paper it could have been

useful to discuss the fact that to sustain a large scale Antarctic glaciations it might requires a cool climate that can allow for Northern Hemisphere ice growth.

We have added explicit mention of this larger ice sheet in the earlier part of the record in L221-224: “We also note that our best-fit modeled AIS sea level contributions are negative (i.e., larger than today) during part of the early Pliocene. Despite larger ice volumes, we simulate WAIS and even EAIS grounding line retreat (Fig. S1), as precipitation outweighs interglacial marine mass loss under this early Pliocene combination of relatively low CO₂ proxy values and invariant insolation (Fig. 2).”

The possibility of Northern Hemisphere ice sheet growth during the Pliocene is a key motivating question for our research. Our ultimate aim is to reconstruct both Southern and Northern Hemisphere transient ice sheet dynamics through the Pliocene in order to investigate the sea level evolution and hemispheric phasing of ice sheet contributions to sea level. This manuscript highlights the Antarctic portion of this ongoing project. We set the stage for future work by adding to L207-209 “These calculations assume that the only Northern Hemisphere source of ice was on Greenland, although future work will investigate potential Northern Hemisphere ice sheet growth.”

4. Erosion and bedrock evolution. It is hard to imagine that the bedrock has remained the same as today over these long timescales. At present there are some deeply incised trough in some areas that might be less marked during the early Pliocene for example. The amount of subglacial sediment may have been different as well. Is the bedrock used the one of the pre-industrial? The bedrock topography is a key driver for ice sheet dynamics and as such these considerations could have been more thoroughly discussed. In the model it would have been possible to use alternative and/or evolving bedrocks to partially quantify this effect?

Yes, this is one potential uncertainty in our approach. Although paleotopographic reconstructions exist, we ultimately chose to use modern topography as initial conditions based on sensitivity testing. L399-403: “Initial conditions for our simulations are provided by a modern ice sheet configuration (DeConto et al., 2021). Pliocene paleotopographic reconstructions deviate only slightly from modern (Paxman et al., 2019; Hochmuth et al., 2020); sensitivity tests conducted using paleotopography produce similar modeled ice sheet fluctuations, but with slightly reduced glacial/interglacial variability, potentially due to the slightly shallower Pliocene subglacial bathymetry and coarser resolution and smoother bed topography of the paleotopographic reconstruction.” Using a modern topography also allowed us to dynamically downscale our continental models to a nested domain (Fig. S2).

Minor comments / questions

L. 370 Which topography the regional climate model use? And how the simulated climate by the RCM is downscaled to the ice sheet model? Using a lapse rate? Is there any precipitation correction?

We varied the GCM/RCM topography, now shown in Fig. S5c. We added to this original sentence "GCM output was then dynamically downscaled to a 60 km resolution over the Antarctic Ice Sheet using a regional climate model (RegCM3) to provide the temperature and precipitation fields passed to the ice sheet model (with temperature and precipitation lapse-rate corrections as needed)."

Fig. M1 The GCM matrix lines for deltaTmixed layer are all below the CCSM4 runs. If I understand correctly it means that the CCSM4 runs are all warmer than the GENESIS runs for this particular variable. Does a direct scaling really make sense then?

The CCSM4 runs are among the warmest Pliocene MIP2 model ensemble members (our GENESIS snapshots fall within the Pliocene MIP2 range; Fig. S6). We choose to use CCSM4 for our scaling methodology for consistency, because it uses the same model as the Liu et al 2009 simulation anchoring the colder end of our climatologies. We discuss this issue in detail in our response to Reviewer 1.

L. 437-441 There is a potential role of Northern Hemisphere ice sheets here that is disregarded in this paper.

This paragraph discusses the absence of any M2 event simulated here. We decided not to speculate further on the lack of an M2 signal in our simulated AIS evolution because (1) the CO₂ reconstruction has a lot of associated uncertainty; and (2) the M2 event has been notoriously difficult to reproduce with ice sheet and climate model simulations (Dolan et al., 2015; 'Modelling the enigmatic Late Pliocene Glacial Event - Marine Isotope Stage M2'). We therefore feel that a statement concluding that a lack of M2 signal in our simulations indicates a Northern Hemisphere M2 contribution would be unfounded.

Technical corrections

L.45 and L.335 The Schoof (2007) formulation of the flux at the grounding line is not really a parametrization. *Corrected.*

L. 341-342 The value given for VCLIFF are in m/yr right? *Corrected.*

L. 376 "glacial cycle" -> deglaciation *Corrected.*

Fig 1. What ESL means here? Volume above flotation in time with respect to the volume above flotation for PI?

Added to Fig. 1 caption: “Simulated Pliocene Antarctic esl (equivalent GMSL contribution relative to modern, calculated from the total ice amount in the domain divided by global ocean area)”

Fig. 1 shows poor quality. *Corrected.*

Reference

Schoof, C. (2007), Ice sheet grounding line dynamics: Steady states, stability, and hysteresis, *J. Geophys. Res.*, 112, F03S28, doi:10.1029/2006JF000664.

Seroussi, H. et al.: ISMIP6 Antarctica: a multi-model ensemble of the Antarctic ice sheet evolution over the 21st century, *The Cryosphere*, 14, 3033–3070, <https://doi.org/10.5194/tc-14-3033-2020>, 2020.

Sun S, Pattyn F, Simon EG, et al. Antarctic ice sheet response to sudden and sustained ice-shelf collapse (ABUMIP). *Journal of Glaciology*. 2020;66(260):891-904. doi:10.1017/jog.2020.67

REVIEWERS' COMMENTS

Reviewer #1 (Remarks to the Author):

The authors have done an excellent job at meeting the review comments on the original manuscript submission. Particularly, the additional supplementary figures really make clear the climatological forcing in these simulations. Figure S6 is an excellent addition, but would be improved with a few technical alterations. It would be helpful to have the axes crossing at the origin, so it is obvious the model is getting colder (I assume that all these temperatures are relative to pre-industrial, but that also needs clarifying). It should also be pointed out that the GENESIS model is a slab ocean model, unlike the PlioMIP models that include a fully coupled ocean atmosphere. If this is not made clear then the GENESIS relationship between SAT and SST looks anomalous. The caption should also make clear that the PlioMIP simulations are modelling isotope stage KM5c (not some generic mid-Pliocene), when the orbital forcing is very close to modern and the Pliocene warming is not at a peak (at least in the benthic oxygen isotope curve). With a few technical alterations to this new figure, I have no hesitation in recommending this manuscript for publication in Nature Communications.

Reviewer #3 (Remarks to the Author):

For this review, the authors have responded clearly and precisely to all my previous comments. I am really grateful for that. I think that the paper can be published in Nature Communications where it will make a valuable contribution to our understanding of past Antarctic ice sheet dynamics.

**Response to second-round reviewer comments on:
“Geologically constrained Antarctic Ice Sheet simulations: 2 million years of
glacial retreat and expansion through the warm Pliocene”
by A. R. Halberstadt et al.**

Reviewer #1:

The authors have done an excellent job at meeting the review comments on the original manuscript submission. Particularly, the additional supplementary figures really make clear the climatological forcing in these simulations. Figure S6 is an excellent addition, but would be improved with a few technical alterations. It would be helpful to have the axes crossing at the origin, so it is obvious the model is getting colder (I assume that all these temperatures are relative to pre-industrial, but that also needs clarifying). It should also be pointed out that the GENESIS model is a slab ocean model, unlike the PlioMIP models that include a fully coupled ocean atmosphere. If this is not made clear then the GENESIS relationship between SAT and SST looks anomalous. The caption should also make clear that the PlioMIP simulations are modelling isotope stage KM5c (not some generic mid-Pliocene), when the orbital forcing is very close to modern and the Pliocene warming is not at a peak (at least in the benthic oxygen isotope curve). With a few technical alterations to this new figure, I have no hesitation in recommending this manuscript for publication in Nature Communications.

Edits to Fig. S6 are shown below.

Figure S6. Climate model snapshots compared to PlioMIP2 ensemble members. Climate model snapshots using the GENESIS GCM with 421 ppm CO₂ produce Pliocene climatologies (surface air temperatures and sea surface temperatures) that fall within the PlioMIP2 ensemble range of variability⁷. Note that the GENESIS GCM has a slab ocean model, while the PlioMIP2 simulations are fully coupled ocean-atmosphere models. Climate model snapshots with lower CO₂ expand below the PlioMIP2 range;

PlioMIP2 simulations were conducted under 400 ppm CO₂ concentrations to reconstruct the KM5c interval (when orbital forcing was close to modern, though Pliocene warming was not peak), while we explore a wider range of glacial/interglacial conditions.

From the Author Checklist ('One final reviewer comment to address in point-by-point response'):

The timing of East Antarctic grounding line retreat at 15 km resolution (Fig. S2, right) is masked. It could be important to show this region because the MICI could play a role there.

The Fig S2 spatial plot in question (reproduced below), rather than masking, shows the two nested domains in color (these domains are restricted in extent and are downscaled using boundary conditions from the continent-wide simulation). The two nested domains shown in (a) and (b) are run at a higher (15km) resolution across the ~2 million year time interval, spanning the Amundsen Sea and the Ross Sea regions. We did not nest any domains over East Antarctica since the goal of this analysis was to investigate any resolution dependence in our simulated grounding-line dynamics. We do not expect the MICI formulation to be sensitive model grid resolution because our parameterization is driven by only by meltwater availability and ice cliff height.

Figure S2. Insensitivity of results to model resolution. Higher-resolution (15km) nested simulations, driven by boundary conditions from a best-fit continental run (40km resolution), produce very similar grounding-line fluctuation patterns (a,b) and extents (c). (a,b) Grounding-line behavior at each timestep is plotted with respect to distance from the continental shelf edge (black star) along the trackline shown in the left-hand-side plot (dashed black line) in the (a) Ross Sea and (b) Amundsen Sea. (c) Nested simulations produce the same grounding line positions through time (right) as the continental simulation (left). Grounding line color indicates the ice sheet grounded area with respect to the modern extent.